# RIDER: 3D RNA Inverse Design with Reinforcement Learning-Guided Diffusion

**Tianmeng Hu**
University of Exeter
tianmeng0824@gmail.com

**Yongzheng Cui**
Central South University
yongzhengcui@csu.edu.cn

**Biao Luo**
Central South University
biao.luo@hotmail.com

**Ke Li**[*]
University of Exeter
k.li@exeter.ac.uk

## ABSTRACT

The inverse design of RNA three-dimensional (3D) structures is crucial for engineering functional RNAs in synthetic biology and therapeutics. While recent deep learning approaches have advanced this field, they are typically optimized and evaluated using native sequence recovery, which is a limited surrogate for structural fidelity, since different sequences can fold into similar 3D structures and high recovery does not necessarily indicate correct folding. To address this limitation, we propose `RIDER`, an RNA Inverse DEsign framework with Reinforcement learning that directly optimizes for 3D structural similarity. First, we develop and pre-train a GNN-based generative diffusion model conditioned on the target 3D structure, achieving a $9\%$ improvement in native sequence recovery over state-of-the-art methods. Then, we fine-tune the model with an improved policy gradient algorithm using four task-specific reward functions based on 3D self-consistency metrics. Experimental results show that `RIDER` improves structural similarity by over $100\%$ across all metrics and discovers designs that are distinct from native sequences.

  github.com/COLA-Laboratory/RIDER

## 1 INTRODUCTION

Ribonucleic acid (RNA) is a fundamental biomolecule with diverse biological functions, such as catalysis, gene regulation, and metabolite sensing (Serganov & Patel, 2007). RNA function is closely linked to its complex three-dimensional (3D) structure (Zhang et al., 2022). The RNA inverse design problem (Hofacker et al., 1994b) – finding a nucleotide sequence that will fold into a desired target structure – is therefore critical for designing RNAs with tailored functions for therapeutics and synthetic biology (Taft et al., 2010; Serganov & Nudler, 2013; Dykstra et al., 2022; Guo, 2010).

RNA structure is hierarchical, progressing from the primary nucleotide sequence to secondary structures formed by local base pairing and culminating in the tertiary 3D structure, which involves long-range interactions, both canonical Watson-Crick and non-canonical pairs. While most computational RNA inverse design target RNA secondary structure (Andronescu et al., 2004; Busch & Backofen, 2006; Garcia-Martin et al., 2013; 2015; Kleinkauf et al., 2015; Eastman et al., 2018; Runge et al., 2019; Zhou et al., 2023), achieving precise biological function often requires control over the final 3D structure, as secondary structure only partially dictates the tertiary fold (Huang et al., 2024). Consequently, recent research has focused on tertiary structure-based inverse design. State-of-the-art methods employ deep learning, using graph neural networks (GNNs) (Joshi et al., 2025; Wong et al., 2024; Tan et al., 2024) or generative diffusion models (Huang et al., 2024). These approaches are inspired by successes in protein design (Dauparas et al., 2022), demonstrate significant improvements in speed and performance over traditional physically-based method (Leman et al., 2020). However,

---

[*]Corresponding author

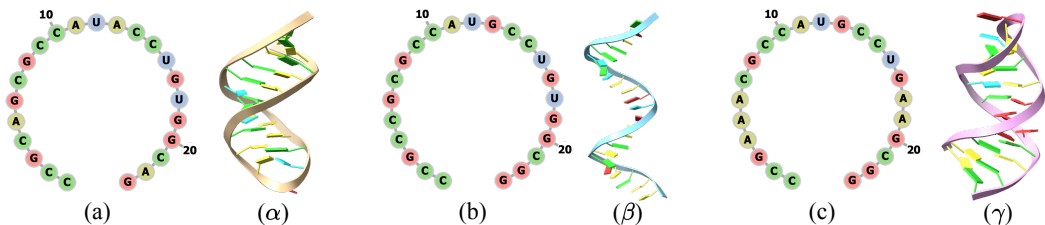

Figure 1: Visualization of sequences (a), (b), and (c), and their corresponding 3D structures ($\alpha$), ($\beta$), and ($\gamma$) predicted by RhoFold (Shen et al., 2024). Although sequences (a) and (b) differ by only 3 nucleotides, and (b) and (c) by 5 nucleotides, their folded structures exhibit clear differences.

they are typically optimized and evaluated based on their ability to recover the native sequence for a given structure, measured by native sequence recovery (NSR) (Joshi et al., 2025).

Optimizing solely for NSR is problematic for RNA design. Unlike proteins, the relationship between RNA sequence and structure is highly degenerate: multiple distinct sequences can fold into similar structures (Assmann et al., 2023). Furthermore, similar sequences do not necessarily result in similar structures, as illustrated in Figure 1. The true objective of RNA inverse design is structural fidelity – finding any sequence that folds correctly – not necessarily recovering the single native sequence observed in the training data. Relying on NSR as a proxy objective has limitations: 1) It does not directly optimize for structural similarity; improving sequence similarity to the native sequence does not ensure the designed structure matches the target (Liu et al., 2025). Notably, most secondary structure design methods avoid NSR, instead optimizing structural distance (Kleinkauf et al., 2015; Eastman et al., 2018; Runge et al., 2019; Zhou et al., 2023). 2) A key problem in RNA biology is to identify RNA sequences with structural similarity to natural sequences (Morandi et al., 2022). Over-optimizing for NSR restricts exploration to sequences near the native one, potentially preventing the discovery of novel, non-native sequences.

To overcome these limitations, we propose RIDER, a novel approach that directly optimizes structural similarity using reinforcement learning (RL), integrating the strengths of generative models with the targeted optimization capabilities of RL. We first pre-train a generative diffusion model on RNA structure datasets, optimizing for NSR to enable the model to learn 3D structural representations and capture sequence-structure relationships. Subsequently, we fine-tune this model using RL, employing a reward function based on the structural similarity between the predicted structure folded from the generated sequence and the target structure. This RL phase allows exploration of the vast sequence space, potentially uncovering novel sequences far from the native one in the dataset. Conceptually, this aligns with recent successes in using RL to fine-tune large generative models for specific objectives (Black et al., 2024).

Our main contributions are:

- We propose the first reinforcement learning framework for RNA 3D inverse design, which directly optimizes structural similarity rather than relying on surrogate sequence recovery objectives.

- We develop and pre-train RIDE, a generative diffusion model for RNA inverse design conditioned on the target structural information, achieving an NSR of $61\%$ and surpassing the strongest baseline at $56\%$.

- We improve existing RL algorithms for fine-tuning diffusion models and apply them to the pre-trained RIDE, resulting in the complete RIDER framework. Specifically, we adopt a batch-mean baseline for advantage estimation and further incorporate a moving average strategy to stabilize training. We design four novel reward functions based on 3D structural self-consistency metrics to guide RL fine-tuning.

- Through experiments, we demonstrate that RIDER achieves over a $100\%$ improvement over existing state-of-the-art methods across all three 3D self-consistency metrics.

## 2 RELATED WORK

**RNA inverse design.** Computational RNA inverse design seeks sequences that fold into a desired target structure. Most existing methods primarily focus on secondary structure. These approaches employ RNA secondary folding prediction tools (Lorenz et al., 2011; Markham & Zuker, 2008; Ali et al., 2023) to compute the structural distance between the designed and target structures, and then search for sequences that meet the design objectives. Methods in this category include local search (Hofacker et al., 1994a; Andronescu et al., 2004; Busch & Backofen, 2006), metaheuristic algorithms (Taneda, 2012; Esmaili-Taheri et al., 2014; Esmaili-Taheri & Ganjtabesh, 2015; Kleinkauf et al., 2015; Zhou et al., 2023), and reinforcement learning (Eastman et al., 2018; Runge et al., 2019). Recent efforts have shifted toward 3D structure design. Physics-based approaches such as Rosetta (Leman et al., 2020) support fixed-backbone design, but are computationally expensive. More recent methods incorporate deep learning: RiboDiffusion (Huang et al., 2024) employs generative diffusion models with Transformer components. RDesign (Tan et al., 2024), RhoDesign (Wong et al., 2024), R3Design (Tan et al., 2025), and gRNAde (Joshi et al., 2025) leverage GNNs to extract 3D structural representations and train generative models to recover sequences from structural inputs, achieving high sequence recovery. A common thread among these state-of-the-art methods is their reliance on supervised learning maximizing sequence recovery, which we argue is an indirect and potentially suboptimal proxy for structural fidelity. Our work addresses this by introducing an RL framework for direct structural optimization.

**RNA structure prediction.** Since experimentally determining RNA structures is labor-intensive and expensive, computational prediction methods have become indispensable (Piao et al., 2017). Early approaches such as RNAComposer (Popenda et al., 2012) assembled structures from libraries of known fragments. More recently, deep learning has become dominant, leveraging statistical patterns learned from existing RNA structures. Representative examples include trRosettaRNA (Wang et al., 2023), DRFold (Li et al., 2023), and RhoFold (Shen et al., 2024). Although RNA structure prediction is essential, progress in RNA functionality and drug discovery ultimately depends on methods for designing novel sequences. Nonetheless, continued improvements in prediction methods are expected to further enhance the effectiveness of our design framework.

**Reinforcement learning.** Reinforcement Learning provides a framework for optimizing policies toward specific goals defined by reward functions and has been widely applied to solve dynamic decision-making problems (Mnih et al., 2015; Lillicrap et al., 2016; Rashid et al., 2020; Hu et al., 2023). RL has achieved remarkable success across a range of challenging domains, including competitive games and multi-agent environments (Lowe et al., 2017; Vinyals et al., 2019), Go (Silver et al., 2016; 2017), robotic control (Haarnoja et al., 2024; Hu & Luo, 2024), and autonomous drone racing (Kaufmann et al., 2023), with several systems attaining or surpassing human-expert performance. Recently, RL has been employed to fine-tune large language models to align their outputs with human intent (Ouyang et al., 2022; Bai et al., 2022), or to enhance reasoning abilities for solving complex mathematical and programming tasks (DeepSeek-AI et al., 2025; Wang et al., 2024). In addition, recent works apply RL to fine-tune pretrained generative models, aligning them with human preferences or optimizing them for specific objectives (Fan et al., 2023; Zhang et al., 2024; Black et al., 2024). For example, DDPO (Black et al., 2024) designs reward functions for text-to-image generation, enabling diffusion models to adapt toward targets such as image compressibility or aesthetic quality.

## 3 METHOD

Our approach for 3D RNA inverse design is formulated as a conditional generative process based on a diffusion model. Given a target RNA 3D backbone structure, the model learns to generate a compatible RNA sequence. The method comprises two main stages: first, a geometric graph representation of the RNA backbone is extracted and processed by a structure encoder to obtain a conditional embedding; second, a diffusion model, conditioned on this structural embedding, iteratively refines a noisy sequence representation to produce the final RNA sequence. Building upon this, we further introduce a reinforcement learning fine-tuning stage to directly optimize the structural similarity of the generated sequences. The pseudocode of the algorithms is provided in Appendix A.

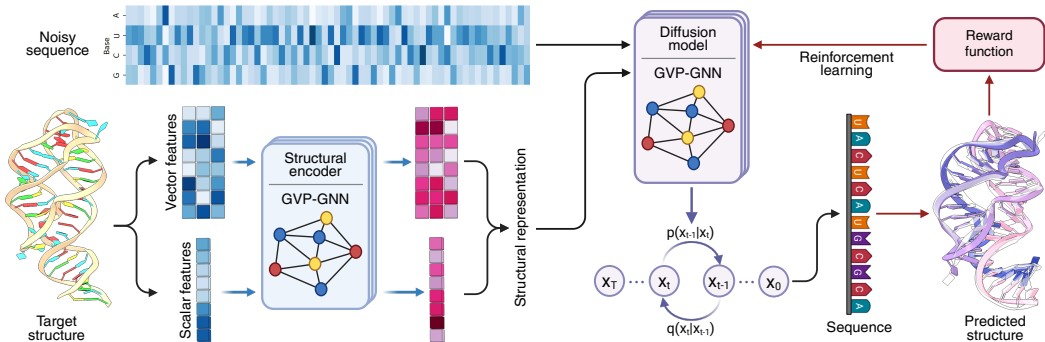

Figure 2: Overview of the RIDER framework. RNA tertiary structures are processed by a GVP-GNN encoder to produce structural embeddings. These embeddings condition the diffusion model for sequence generation, which is further optimized by RL to maximize structural similarity.

## 3.1 TERTIARY STRUCTURE REPRESENTATION

To capture the geometric properties of RNA tertiary structures, we represent each molecule as a geometric graph, where nodes correspond to nucleotides and edges encode spatial proximity. A structural encoder based on GVP-GNN (Jing et al., 2021) processes this graph to produce node-level embeddings that are equivariant to 3D rotations and translations. These embeddings compactly summarize the local and global geometry of the RNA backbone and serve as conditioning context for the diffusion model. Details of the structural encoder are provided in Appendix B.1. Figure 2 illustrates the overall pipeline.

## 3.2 CONDITIONAL DIFFUSION MODEL FOR RNA SEQUENCE GENERATION

We formulate RNA sequence design as learning a conditional distribution $p(\mathbf{x}_0 \mid \mathbf{h}_c)$, where $\mathbf{x}_0$ denotes the one-hot encoded sequence ($N$ nucleotides, $C = 4$ bases, $\mathbf{x}_0 \in \{0, 1\}^{N \times C}$) and $\mathbf{h}_c$ is the structural embedding from the encoder. To model this distribution, we adopt a variance-preserving diffusion probabilistic model (Ho et al., 2020; Song et al., 2021b), which learns to predict the noise added to a clean sequence. The forward diffusion process gradually adds Gaussian noise to the clean data $\mathbf{x}_0$ over a continuous time variable $t \in [\epsilon_{\texttt{time}}, T]$, where $\epsilon_{\texttt{time}}$ is a small positive constant and $T$ is the final time. The noised sample $\mathbf{x}_t$ at time $t$ is given by:

$$\mathbf{x}_t = \alpha_t \mathbf{x}_0 + \sigma_t \varepsilon, \tag{1}$$

where $\varepsilon \sim \mathcal{N}(0, I)$, and $\alpha_t, \sigma_t$ are schedule functions derived from a predefined noise schedule $\beta_t$. Specifically, $\alpha_t^2 = \exp\left(-\int_0^t \beta_s ds\right)$ and $\sigma_t^2 = 1 - \alpha_t^2$. The noise level for a given $t$ is often represented as $\lambda_t = \log(\alpha_t^2/\sigma_t^2)$.

The core of our generative model is a noise prediction network $\epsilon_\theta(\mathbf{x}_t, t, \mathbf{h}_c)$ with parameters $\theta$. This network is trained to predict the noise $\varepsilon$ from the noisy input $\mathbf{x}_t$, conditioned on the time $t$ and the structural context $\mathbf{h}_c$. The network $\epsilon_\theta$ is composed of $L_D = 5$ GVP-GNN layers. The input to this noise prediction network for each node $i$ combines its noisy sequence representation $(\mathbf{x}_t)_i$, an embedding of the time step $t_{\texttt{emb}}$, and its structural embedding $(\mathbf{h}_c)_i$. This combined representation is then processed by the network to predict the noise component $(\hat{\varepsilon})_i$ for each nucleotide. The model is trained by minimizing the mean squared error between the true noise $\varepsilon$ and the predicted noise $\hat{\varepsilon}_\theta$:

$$\mathcal{L}_{\texttt{pretrain}}(\theta) = \mathbb{E}_{t, \mathbf{x}_0, \varepsilon, \mathbf{h}_c}\left[||\varepsilon - \epsilon_\theta(\alpha_t \mathbf{x}_0 + \sigma_t \varepsilon, t, \mathbf{h}_c)||^2\right]. \tag{2}$$

During inference, we generate sequences by reversing the diffusion process. Starting from a random Gaussian noise sample $\mathbf{x}_T \sim \mathcal{N}(0, I)$, we iteratively denoise it using the Denoising Diffusion Implicit Models (DDIM) (Song et al., 2021a) sampler. After $N_{\texttt{steps}}$ iterations, we obtain an estimate of the clean sequence $\hat{\mathbf{x}}_0$. The final discrete RNA sequence is obtained by applying an argmax operation over the $C$ channels for each nucleotide in $\hat{\mathbf{x}}_0$.

### 3.3 REINFORCEMENT LEARNING FOR STRUCTURAL SIMILARITY OPTIMIZATION

Although the pre-trained diffusion model can learn sequence-structure relationships, its training objective does not directly maximize the structural similarity between the folded structure of the generated sequence and the target structure. To overcome this limitation, we employ a reinforcement learning-based method to fine-tune the pre-trained diffusion model, inspired by the denoising diffusion policy optimization (DDPO) framework (Black et al., 2024), to directly optimize for rewards related to downstream tasks.

We frame the denoising sampling process of the diffusion model as a Markov decision process (MDP). In this MDP:

- **State** $s_t$: Defined as $(\mathbf{x}_t, t, \mathbf{h}_c)$, representing the current noisy sequence, time step, and structural condition.
- **Action** $a_t$: Defined as the transition from $\mathbf{x}_t$ to $\mathbf{x}_{t-\Delta t}$ in one denoising step. In our context, this corresponds to the noise (or equivalently, the predicted $\hat{\mathbf{x}}_0$) predicted by the model $\epsilon_\theta$ given $s_t$, which determines the mean of the next state $\mathbf{x}_{t-\Delta t}$.
- **Policy** $\pi_\theta(a_t|s_t)$: Parameterized by the diffusion model $\epsilon_\theta$.
- **Reward** $R(\hat{\mathbf{x}}_0, \mathbf{h}_c^{\texttt{target}})$: Obtained only at the end of the trajectory (i.e., after generating the complete $\hat{\mathbf{x}}_0$).

**Advantage estimation**  We employ a policy gradient method to fine-tune the diffusion model. The core idea is to update the model parameters $\theta$ using the complete denoising trajectories $\{\mathbf{x}_T, \mathbf{x}_{T-1}, ..., \hat{\mathbf{x}}_0\}$ and their final rewards $R_{traj} = R(\hat{\mathbf{x}}_0, \mathbf{h}_c^{\texttt{target}})$. The original DDPO applies an importance sampling (IS) based estimator:

$$\nabla_\theta \mathcal{J}_{\text{DDRL}} = \mathbb{E}_{\tau \sim p(\tau|\pi_{\theta_{\text{old}}})} \left[ \sum_{k=0}^{N_{\text{steps}}-1} \frac{\pi_\theta(a_k|s_k)}{\pi_{\theta_{\text{old}}}(a_k|s_k)} \nabla_\theta \log \pi_\theta(a_k|s_k) R_{\texttt{traj}} \right], \qquad (3)$$

where $a_k$ represents the action of transitioning from state $s_k$ (i.e., $(\mathbf{x}_{t_k}, t_k, \mathbf{h}_c)$) to $\mathbf{x}_{t_{k-1}}$ at time step $k$, $\pi_\theta(a_k|s_k)$ is the probability of this action under the current policy, and $\pi_{\theta_{\text{old}}}$ is the old policy used for sampling. While this approach is effective in image generation, we found that directly using the reward $R_{\texttt{traj}}$ in our RNA design context leads to high variance and instability during optimization. Therefore, we replace the reward function with an advantage function $A(\hat{\mathbf{x}}_0, \mathbf{h}_c^{\texttt{target}})$ to reduce the variance of gradient estimates. Specifically, for each target structure, we collect a batch of experience trajectories by running the denoising sampling process multiple times. Each trajectory includes a sequence of latent states $\mathbf{x}_t$, the log-probability $\log \pi_{\theta_{\text{old}}}(a_t|s_t)$ for each step, and the final structural similarity reward $R_{\texttt{traj}}$. We then compute the average reward over this batch as a baseline $b$:

$$b = \mathbb{E}_{\tau \sim p(\tau|\pi_{\theta_{\text{old}}})}[R_{\texttt{traj}}]. \qquad (4)$$

However, we observe in our experiments that using a baseline based solely on the batch average can lead to instability. Due to the stochasticity of the sampling process, baseline values computed from adjacent batches may vary significantly, resulting in large fluctuations in advantage estimates and hindering stable learning. Increasing the batch size may mitigate this issue but at the cost of reduced training efficiency. To address this, we adopt a moving average strategy for baseline estimation. Let $b^{(i)}$ denote the moving average baseline after the $i$-th batch of experience collection, and let $\bar{R}_{\texttt{batch}}^{(i)}$ be the average reward of the $i$-th batch. The baseline is updated according to:

$$b^{(i)} = \beta_{\texttt{baseline}} \cdot b^{(i-1)} + (1 - \beta_{\texttt{baseline}}) \cdot \bar{R}_{\texttt{batch}}^{(i)}, \qquad (5)$$

where $b^{(0)}$ is initialized as the average reward of the first batch, and $\beta_{\text{baseline}}$ is a smoothing factor that controls the update rate. The smoothed baseline $b \equiv b^{(i)}$ is then used in advantage computation[1]. The advantage function is then defined as $A(\hat{\mathbf{x}}_0, \mathbf{h}_c^{\texttt{target}}) = R_{\texttt{traj}} - b$, and the policy gradient estimator based on advantage is correspondingly modified to:

$$\nabla_\theta \mathcal{J}_{\text{RL}} = \mathbb{E}_{\tau \sim p(\tau|\pi_{\theta_{\text{old}}})} \left[ \left( \sum_{k=0}^{N_{\text{steps}}-1} \frac{\pi_\theta(a_k|s_k)}{\pi_{\theta_{\text{old}}}(a_k|s_k)} \nabla_\theta \log \pi_\theta(a_k|s_k) \right) A(\hat{\mathbf{x}}_0, \mathbf{h}_c^{\texttt{target}}) \right]. \qquad (6)$$

---

[1]Appendix D.1 investigates the impact of advantage function estimation.

During the sampling process, to ensure diversity in each batch of samples and effectively explore the policy space, we perform one deterministic sampling (temperature parameter set to 0) and multiple stochastic samplings. In the stochastic sampling phase, we uniformly sample a temperature from a discrete set (e.g., $\{0.1, 0.3, 0.5, 0.7, 0.9\}$ with a probability of 1/5 for each) to control the randomness of the generation process. Furthermore, we adopt the clipping technique (Schulman et al., 2017) to constrain the policy updates. We apply the importance sampling ratio $r_k(\theta) = \frac{\pi_\theta(a_k|s_k)}{\pi_{\theta_{\text{old}}}(a_k|s_k)}$ from Equation (6) to the advantage function and clip the result, forming the following optimization objective:

$$\mathcal{L}^{RL}(\theta) = \mathbb{E}_{\tau \sim p(\tau|\pi_{\theta_{\text{old}}})} \left[ \sum_{k=0}^{N_{\text{steps}}-1} \min(r_k(\theta)A, \text{clip}(r_k(\theta), 1 - \epsilon_{\text{clip}}, 1 + \epsilon_{\text{clip}})A) \right], \quad (7)$$

where $A$ is the advantage estimation of the entire trajectory, and $\epsilon_{\text{clip}}$ is the clipping parameter.

**Reward function**   The reward function is designed to quantify the structural similarity between the predicted 3D structure—obtained by folding the generated RNA sequence—and the target 3D structure. We use the computational structure prediction model RhoFold (Shen et al., 2024) to predict the folded structure of the generated sequence. We employ three widely used structural similarity metrics:

- **GDT_TS** (Zemla et al., 1999) (Global Distance Test Total Score): measures the percentage of backbone atoms within preset distance thresholds (1Å, 2Å, 4Å, 8Å) of their counterparts after optimal superposition. It emphasizes well-aligned regions and ranges from 0 to 1, with higher scores indicating better structural alignment.
- **RMSD** (Kabsch, 1976) (Root Mean Square Deviation): computes the average displacement between corresponding backbone atoms after superposition. It is sensitive to outliers; lower values indicate higher similarity.
- **TM-score** (Zhang & Skolnick, 2004) (Template Modeling Score): evaluates global fold similarity using a length-normalized function. It is tolerant to local deviations and suitable for comparing structures of different lengths. Values range from 0 to 1, with higher scores reflecting better agreement.

A detailed explanation of these metrics is provided in Appendix C. Based on these metrics, we explore various reward functions. First, we define base reward functions using individual metrics to analyze the model's ability to optimize for specific structural features:

$$R^{\text{gdt}} = (\text{GDT\_TS} \times w_{\text{gdt\_scale}})^2, \quad (8)$$

$$R^{\text{tm}} = (\text{TM-score} \times w_{\text{tm\_scale}})^2, \quad (9)$$

$$R^{\text{rmsd}} = -(\text{RMSD} \times w_{\text{rmsd\_scale}})^2, \quad (10)$$

where $w_{\text{gdt\_scale}}$, $w_{\text{tm\_scale}}$, and $w_{\text{rmsd\_scale}}$ are scaling factors. Considering that RMSD and GDT_TS reflect structural similarity from different perspectives, we also design a combined reward function to balance the pursuit of precise atomic matching and overall fold similarity:

$$R^{\text{gdt\_rmsd}} = -(\text{RMSD} \times w_{\text{rmsd\_scale}})^2 + (\text{GDT\_TS} \times w_{\text{gdt\_scale}})^2. \quad (11)$$

It is considered that when GDT_TS $> 0.5$ or RMSD $< 2.0$Å, the predicted structure has significant similarity to the target structure, meeting an acceptable design standard (Tan et al., 2024; Joshi et al., 2025). To encourage the model to generate sequences with high structural fidelity, we design an additional reward component $R_{\text{bonus}}$ for sequences that meet these criteria, computed as:

$$R_{\text{bonus}} = \begin{cases} (\text{GDT\_TS} - \tau_{\text{gdt}}) \times w_{\text{bonus\_gdt}}, & \text{if GDT\_TS} > \tau_{\text{gdt}} \\ (\tau_{\text{rmsd}} - \text{RMSD}) \times w_{\text{bonus\_rmsd}}, & \text{else if RMSD} < \tau_{\text{rmsd}} \\ 0, & \text{otherwise} \end{cases} \quad (12)$$

where $\tau_{\text{gdt}} = 0.5$ and $\tau_{\text{rmsd}} = 2.0$ are threshold values, and $w_{\text{bonus\_gdt}}$ and $w_{\text{bonus\_rmsd}}$ are reward scaling coefficients. The final total reward function $R_{\text{final}}$ used in our experiments is defined as the sum of a base reward and a bonus reward:

$$R_{\text{final}} = R_{\text{base}} + R_{\text{bonus}}, \quad (13)$$

where $R_{\text{base}}$ corresponds to one of the four base reward functions defined above.

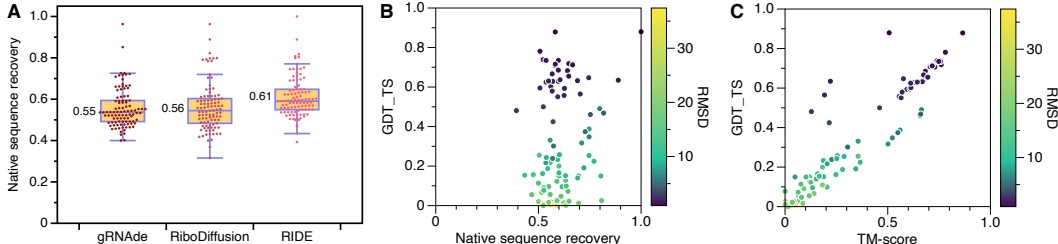

Figure 3: Results of supervised learning pre-training. **A**. Comparison of native sequence recovery on the test set. RIDE is compared against RiboDiffusion and gRNAde. The best NSR among 16 designs per target (sampled at temperature 0.1) is reported. **B**. Relationship between NSR and structural similarity (GDT_TS and RMSD) for RIDE designs. Color denotes RMSD. **C**. Correlation among GDT_TS, TM-score, and RMSD for the designed structures.

## 4 EXPERIMENTS

### 4.1 SETTINGS

**Dataset** We use the RNA tertiary structure dataset published by Joshi et al. (2025) for pre-training and evaluating our model. The dataset is derived from the RNASolo repository (Adamczyk et al., 2022) and contains $4,223$ unique RNA sequences and a total of $12,011$ RNA structures. It is pre-partitioned based on structural similarity into a training set, a validation set, and a test set, with the latter two each containing 100 samples.

**Hyperparameters** Both the structure encoder and the noise prediction network in the diffusion model use 5 layers of GVP-GNNs. The initial learning rate is set to $3 \times 10^{-4}$ and is decayed by a factor of $0.9$ if the validation performance does not improve for 5 consecutive epochs. The model is trained for a total of 150 epochs. The overall number of parameters in our model is 10.2 million. All GVP-GNN layers apply a dropout rate of $0.5$ to mitigate overfitting. To improve robustness, Gaussian noise with a standard deviation of $0.1$ is added to the node coordinates during training. For sequence generation, we use the DDIM sampler (Song et al., 2021a) with $N_{\text{steps}} = 50$ denoising steps. After pre-training, we fine-tune the diffusion model using a policy gradient algorithm. The learning rate is set to $5 \times 10^{-5}$ without learning rate scheduling. Training is conducted for 80 reinforcement learning epochs, with 2 policy updates performed in each epoch. The batch size is 60, meaning that 60 experience trajectories (each corresponding to a full denoising process) are sampled per epoch. Appendix B provides detailed information on the model architecture and training hyperparameters.

### 4.2 SUPERVISED LEARNING PRE-TRAINING

We compare our proposed conditional diffusion model RIDE with two recent state-of-the-art (SOTA) methods: RiboDiffusion (Huang et al., 2024) and gRNAde (Joshi et al., 2025). RiboDiffusion is a Transformer-based diffusion model, whereas gRNAde is a GNN-based generative model that employs an autoregressive decoder to generate RNA sequences.

**RIDE achieves the best sequence recovery** We evaluate the three methods on the test set by comparing their NSR. For each target structure, each method performs 16 sampling runs at a temperature of 0.1 to generate 16 candidate sequences, and the best NSR among them is recorded. The results are shown in Figure 3A. RIDE achieves an average sequence recovery of $61\%$, representing improvements of $9\%$ and $11\%$ over RiboDiffusion and gRNAde, respectively. These results demonstrate that, even without reinforcement learning fine-tuning, our method outperforms existing SOTA baselines in terms of sequence recovery.

**Native sequence recovery does not reflect structural similarity** Using the same methodology as in Figure 3A, we employ RIDE to perform inverse design on the 100 structures in the test set. For each structure, we record the best native sequence recovery and its corresponding 3D structural

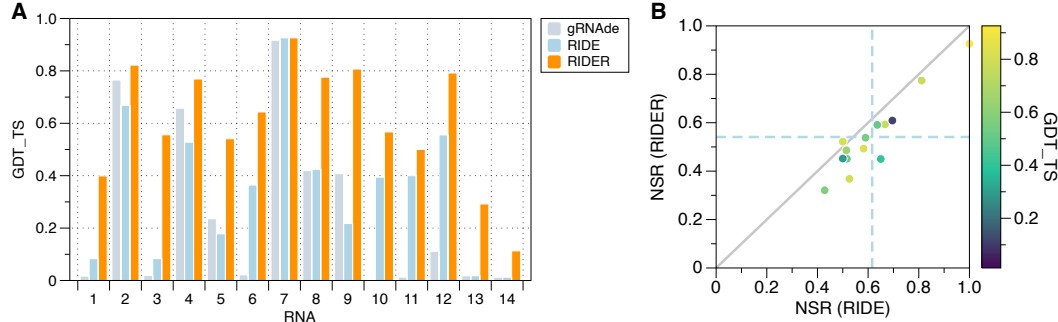

Figure 4: Results of reinforcement learning fine-tuning. **A**. GDT_TS comparison on 14 RNA structures of interest (Das et al., 2010) for gRNAde, `RIDE` (pre-trained), and `RIDER` (fine-tuned with $R^{\text{gdt\_rmsd}}$). **B**. Comparison of native sequence recovery before (`RIDE`) and after (`RIDER`) RL fine-tuning. Color indicates GDT_TS after RL fine-tuning. The results for the other two metrics are provided in Appendix D.4.

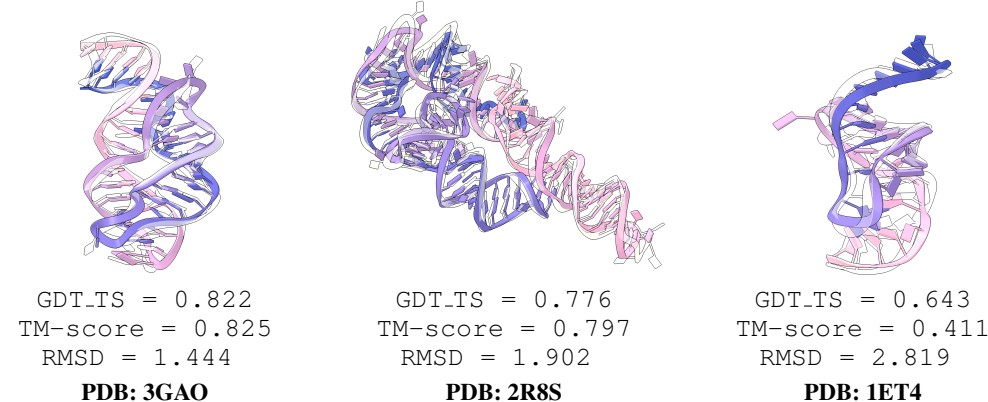

| GDT_TS = 0.822 | GDT_TS = 0.776 | GDT_TS = 0.643 |
| TM-score = 0.825 | TM-score = 0.797 | TM-score = 0.411 |
| RMSD = 1.444 | RMSD = 1.902 | RMSD = 2.819 |
| **PDB: 3GAO** | **PDB: 2R8S** | **PDB: 1ET4** |

Figure 5: Visualization of designed examples. Structures folded from sequences generated by `RIDER` are shown in color, while the target structures are shown in semi-transparent yellow.

similarity metrics, as shown in Figure 3B. The results reveal no clear correlation between sequence recovery and 3D structural similarity. For example, when sequence recovery is around $50\%$, the associated GDT_TS values vary widely from 0 to 0.9. As sequence recovery increases, the distribution of GDT_TS does not consistently shift toward higher values. In fact, even at high levels of sequence recovery, the resulting structures may still exhibit low similarity to the target structure. These observations suggest that native sequence recovery alone is insufficient to assess the structural fidelity of designed sequences. This further highlights that optimizing sequence recovery is not an ideal surrogate objective for 3D RNA inverse design.

**Relationship among three self-consistency metrics** Figure 3C illustrates the relationships among GDT_TS, TM-score, and RMSD. GDT_TS and TM-score show a strong correlation, with a Pearson correlation coefficient of $0.885$. A few outliers highlight differences between these metrics. For instance, some samples exhibit low RMSD values (indicated by darker colors), high GDT_TS, but unexpectedly low TM-scores. This discrepancy arises because TM-score is more sensitive to deviations in shorter structures. Since these three metrics capture structural similarity from different perspectives, we design distinct reward functions based on each of them during the RL fine-tuning phase to investigate their respective impact on performance.

## 4.3 REINFORCEMENT LEARNING FINE-TUNING

After pre-training, we further fine-tune `RIDE` using reinforcement learning, denoted as `RIDER`.

Table 1: Performance comparison of different methods and reward functions on the test set. Results are reported as the mean and standard deviation over 5 independent runs. Percentages in parentheses indicate the proportion of designs that meet the structural designability thresholds (GDT_TS $\geq 0.5$, TM-score $\geq 0.45$, RMSD $\leq 2$Å).

| Method | Reward function | 3D self-consistency metrics | | | NSR |
|---|---|---|---|---|---|
| | | GDT_TS $\uparrow$ | RMSD $\downarrow$ | TM-score $\uparrow$ | |
| RIDER | $R^{\mathtt{tm}}$ | $\mathbf{0.62 \pm 0.04}(\mathbf{72\%})$ | $4.31 \pm 0.70(31\%)$ | $\mathbf{0.61 \pm 0.03}(\mathbf{72\%})$ | $0.45 \pm 0.04$ |
| | $R^{\mathtt{gdt}}$ | $0.58 \pm 0.05(63\%)$ | $4.60 \pm 0.55(30\%)$ | $0.56 \pm 0.05(64\%)$ | $0.47 \pm 0.06$ |
| | $R^{\mathtt{rmsd}}$ | $0.56 \pm 0.03(60\%)$ | $4.09 \pm 0.40(31\%)$ | $0.55 \pm 0.06(64\%)$ | $0.49 \pm 0.05$ |
| | $R^{\mathtt{gdt\_rmsd}}$ | $\mathbf{0.62 \pm 0.02}(\mathbf{72\%})$ | $\mathbf{3.35 \pm 0.44}(\mathbf{33\%})$ | $0.56 \pm 0.03(68\%)$ | $0.47 \pm 0.05$ |
| RIDE | – | $0.33 \pm 0.03(31\%)$ | $10.36 \pm 0.19(8\%)$ | $0.33 \pm 0.03(36\%)$ | $\mathbf{0.61 \pm 0.02}$ |
| RIDE (Best-of-N) | – | $0.42 \pm 0.06(46\%)$ | $8.63 \pm 0.90(15\%)$ | $0.39 \pm 0.05(45\%)$ | $0.50 \pm 0.06$ |
| RiboDiffusion | – | $0.13 \pm 0.02(5\%)$ | $15.68 \pm 0.26(1\%)$ | $0.14 \pm 0.03(6\%)$ | $0.56 \pm 0.04$ |
| gRNAde | – | $0.28 \pm 0.06(27\%)$ | $10.89 \pm 0.61(3\%)$ | $0.30 \pm 0.05(28\%)$ | $0.55 \pm 0.03$ |

Table 2: Cross-predictor validation using the independent AlphaFold3 oracle.

| Model | Reward function | GDT_TS $\uparrow$ | RMSD $\downarrow$ | TM-score $\uparrow$ |
|---|---|---|---|---|
| RIDER (AlphaFold3) | $R^{\mathtt{tm}}$ | $0.56 \pm 0.07(64\%)$ | $4.69 \pm 0.79(30\%)$ | $0.56 \pm 0.06(65\%)$ |
| RIDER (AlphaFold3) | $R^{\mathtt{gdt\_rmsd}}$ | $\mathbf{0.57 \pm 0.06}(\mathbf{65\%})$ | $\mathbf{3.80 \pm 0.61}(\mathbf{32\%})$ | $\mathbf{0.56 \pm 0.05}(\mathbf{67\%})$ |
| gRNAde (AlphaFold3) | – | $0.26 \pm 0.04(26\%)$ | $9.96 \pm 0.77(3\%)$ | $0.28 \pm 0.04(26\%)$ |

**RIDER achieves substantial improvements in 3D structural similarity**   Figure 4A shows results on 14 RNA structures of interest identified by Das et al. (2010), all of which are included in the test set. RIDER delivers significant improvements on the vast majority of these structures, with more than 100% improvement on half of them. Table 1 summarizes the complete results across the test set. Across all three 3D self-consistency metrics, RIDER achieves more than 100% improvement relative to the best-performing baseline. Using GDT_TS as the evaluation criterion, 72% of RIDER designs exceed the threshold of 0.5, compared to only 27% for gRNAde. In terms of RMSD, 33% of RIDER sequences achieve $\leq 2$Å, compared to only 3% for gRNAde. Figure 5 illustrates three representative examples of designed structures.

**RIDER can discover high-quality designs different from native sequences**   Figure 4B shows the change in NSR before and after RL fine-tuning. The color of each point denotes the GDT_TS value after fine-tuning. We observe that, in most cases, the NSR of RIDER designs is lower than that of RIDE, yet these sequences achieve higher GDT_TS.

**RIDER generalizes across different oracles**   To further evaluate the generalization capability of RIDER models, we design a variant using AlphaFold3 (Abramson et al., 2024) to replace the original RhoFold. From the results in Table 2, we find that RIDER maintains its superiority under this cross-oracle evaluation. In particular, there are minor performance differences when using different oracles: with the $R^{\mathtt{gdt\_rmsd}}$ reward, our GDT_TS score is 0.57 with AlphaFold3, which is more than doubled (+119%) the score of the baseline (0.26). These results demonstrate that RIDER captures generalizable principles of RNA design rather than overfitting a single predictor.

**Impact of reward functions**   Table 1 reports the impact of the four reward functions introduced in Section 3.3. The TM-score-based reward $R^{\mathtt{tm}}$ performs strongly on both TM-score and GDT_TS, though it is slightly less effective on RMSD. In contrast, $R^{\mathtt{rmsd}}$ and $R^{\mathtt{gdt}}$ yield weaker results when used individually. Notably, the composite reward $R^{\mathtt{gdt\_rmsd}}$, which combines GDT_TS and RMSD, achieves the most balanced performance overall. We attribute this to the complementary nature of the two metrics, which jointly capture both local atomic accuracy and global fold similarity. Appendix D.2 provides additional evaluation results for all four reward functions.

## 5 CONCLUSION

We propose a novel two-stage framework for 3D RNA inverse design that directly optimizes structural similarity. In the first stage, we develop `RIDE`, a conditional diffusion model trained via supervised learning that surpasses SOTA methods in native sequence recovery. In the second stage, we develop `RIDER`, which significantly improves the similarity between designed and target structures through reinforcement learning. Future work may extend this framework with multi-objective rewards and explore experimental validation of the generated sequences.

## 6 ACKNOWLEDGMENTS

This work was supported by the UKRI Future Leaders Fellowship under Grant MR/S017062/1 and MR/X011135/1; in part by National Natural Science Foundation of China under Grant 62373375, U2341216, 62376056 and 62076056; in part by the Science and Technology Innovation Program of Hunan Province under Grant 2024RC1011 and the Natural Science Foundation of Hunan Province under Grant 2025JJ10007; in part by the Isambard-AI, Royal Society Faraday Discovery Fellowship (FDF/S2/251014), BBSRC Transformative Research Technologies (UKRI1875), Royal Society International Exchanges Award (IES/R3/243136), Kan Tong Po Fellowship (KTP/R1/231017); and the Amazon Research Award and Alan Turing Fellowship.

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

## A  ALGORITHM PSEUDOCODE

This section provides detailed pseudocode for the key algorithmic components of our proposed method. Specifically, we include three main procedures:

- **Algorithm 1** describes the supervised pre-training stage of the conditional diffusion model (`RIDE`), where the model learns to predict noise added to clean RNA sequences given a target 3D structure.

- **Algorithm 2** presents the DDIM-based sampling procedure for generating RNA sequences conditioned on a target structure using the pre-trained model. This forms the basis of both inference and experience collection in the RL stage.

- **Algorithm 3** outlines the reinforcement learning fine-tuning stage (`RIDER`), where a policy gradient method is used to optimize the diffusion model toward improved structural fidelity based on 3D evaluation metrics (GDT_TS, TM-score, RMSD).

Together, these algorithms define the full learning pipeline of our framework.

---

**Algorithm 1** `RIDE`: Pre-training of Conditional Diffusion Model

---

**Require:** Training dataset $\mathcal{D} = \{(y_0^{(i)}, X_{\text{target}}^{(i)})\}$, number of diffusion steps $T_{\text{diff}}$, model parameters $\theta$
**Require:** Structure encoder $E_{\text{gnn}}$, GVP-GNN noise prediction network $\epsilon_\theta$
**Ensure:** Pre-trained model parameters $\theta$
1: Initialize model parameters $\theta$ (for $\epsilon_\theta$)
2: **for** each training iteration **do**
3:   Sample a minibatch $(y_0, X_{\text{target}})$ from $\mathcal{D}$
4:   Compute structure condition $\mathcal{C} = E_{\text{gnn}}(X_{\text{target}})$
5:   Sample timestep $t \sim \text{Uniform}(\{1, \ldots, T_{\text{diff}}\})$
6:   Sample standard Gaussian noise $\varepsilon \sim \mathcal{N}(0, I)$
7:   Generate noisy sequence $y_t = \alpha_t y_0 + \sigma_t \varepsilon$ using the forward process
8:   Predict noise using the model: $\hat{\varepsilon} = \epsilon_\theta(y_t, t, \mathcal{C})$
9:   Compute pre-training loss: $L_{\text{pretrain}} = \|\varepsilon - \hat{\varepsilon}\|^2$
10:   Compute gradients $\nabla_\theta L_{\text{pretrain}}$
11:   Update parameters: $\theta \leftarrow \theta - \eta \nabla_\theta L_{\text{pretrain}}$
12: **end for**
13:
14: **return** Trained model parameters $\theta$

---

**Algorithm 2** Sequence Sampling via DDIM

---

**Require:** Target structure $X_{\text{target}}$, trained or fine-tuned model parameters $\theta$ (for $\epsilon_\theta$)
**Require:** Structure encoder $E_{\text{gnn}}$, GVP-GNN noise prediction network $\epsilon_\theta$
**Require:** Total number of sampling steps $N_{\text{steps}}$
**Ensure:** Designed RNA sequence $\hat{y}_0$
1: Compute structure condition $\mathcal{C} = E_{\text{gnn}}(X_{\text{target}})$
2: Sample initial noise vector $y_{N_{\text{steps}}} \sim \mathcal{N}(0, I)$
3: **for** $k = N_{\text{steps}}$ **down to** 1 **do**
4:   Determine current and previous time steps $t_k$ and $t_{k-1}$
5:   Predict noise: $\hat{\varepsilon}_k = \epsilon_\theta(y_{t_k}, t_k, \mathcal{C})$
6:   Compute $y_{t_{k-1}}$ using DDIM update rule with $y_{t_k}$ and $\hat{\varepsilon}_k$
7: **end for**
8: Set $\hat{y}_0 \leftarrow y_0$
9: Obtain discrete sequence: $\hat{y}_{0,\text{discrete}} = \text{argmax}(\hat{y}_0)$
10:
11: **return** $\hat{y}_{0,\text{discrete}}$

---

---

**Algorithm 3** `RIDER`: Reinforcement Learning Fine-Tuning

---

**Require:** Pretrained policy network $\pi_\theta$; target structure dataset $\mathcal{D}_{\text{target}}$; structure encoder $E_{\text{gnn}}$; RNA structure predictor $F_{\text{pred}}$; reward function $R$

**Require:** Clipping coefficient $\epsilon_{\text{clip}}$; learning rate $\eta$; number of RL epochs $N$; number of trajectories per epoch $M_{\text{traj}}$; number of policy update steps per epoch $K_{\text{update}}$

**Ensure:** Fine-tuned policy parameters $\theta$

1: **for** $epoch = 1$ **to** $N$ **do**
2:     $\theta_{\text{old}} \leftarrow \theta$
3:     Initialize experience buffer $\mathcal{B}$
4:     **for** $m = 1$ **to** $M_{\text{traj}}$ **do**
5:         Sample target structure $X_{\text{target}}$ from $\mathcal{D}_{\text{target}}$
6:         Compute structure condition $\mathcal{C} = E_{\text{gnn}}(X_{\text{target}})$
7:         $(\hat{y}_0, \text{trajectory\_data}_m) \leftarrow$ sampling
8:         $X_{\text{pred}} = F_{\text{pred}}(\hat{y}_0)$
9:         $R_{\text{traj}}^{(m)} = R(X_{\text{pred}}, X_{\text{target}})$
10:        Store $(\text{trajectory\_data}_m, R_{\text{traj}}^{(m)})$ in $\mathcal{B}$
11:     **end for**
12:     Compute baseline $b$
13:     **for** $n = 1$ **to** $K_{\text{update}}$ **do**
14:         **for** each $(\text{trajectory\_data}, R_{\text{traj}})$ in $\mathcal{B}$ **do**
15:            Compute advantage: $A = R_{\text{traj}} - b$
16:           **for** each $(s_k, a_k, \log \pi_{\theta_{\text{old}}}(a_k|s_k))$ in trajectory **do**
17:              Compute $\log \pi_\theta(a_k|s_k)$
18:              Compute importance ratio $r_k(\theta) = \exp(\log \pi_\theta(a_k|s_k) - \log \pi_{\theta_{\text{old}}}(a_k|s_k))$
19:              $L_{\text{traj}} \leftarrow L_{\text{traj}} + \min(r_k(\theta)A, \text{clip}(r_k(\theta), 1 - \epsilon_{\text{clip}}, 1 + \epsilon_{\text{clip}})A)$
20:           **end for**
21:         **end for**
22:         $L^{\text{RL}} \leftarrow L_{\text{traj}}/M_{\text{traj}}$
23:         Compute gradient $\nabla_\theta L^{\text{RL}}$
24:         Update parameters: $\theta \leftarrow \theta - \eta \nabla_\theta L^{\text{RL}}$
25:     **end for**
26: **end for**
27:
28: **return** Fine-tuned parameters $\theta$

---

## B  MODEL ARCHITECTURE AND TRAINING HYPERPARAMETERS

This section provides detailed descriptions of the `RIDE` architecture, pre-training and fine-tuning hyperparameters, as well as the computational setup used during training. Figure 6 illustrates the overall architecture and learning pipeline of our proposed method. The framework consists of two stages: (1) `RIDE`, a conditional diffusion model that is pre-trained to generate RNA sequences from target 3D structures, and (2) `RIDER`, a reinforcement learning fine-tuning stage that optimizes the diffusion model for structural fidelity using 3D self-consistency metrics (GDT_TS, TM-score, RMSD). The model takes as input a 3D structure embedding and iteratively denoises a sequence initialized from Gaussian noise, guided by a multi-layer GVP-GNN encoder-decoder architecture. The predicted sequences are evaluated via a structure prediction model, and reward signals are computed to fine-tune the policy.

### B.1  STRUCTURE REPRESENTATION

To effectively capture the geometric intricacies of RNA tertiary structures, we first convert the input backbone into a graph representation, which is then processed by a Geometric Vector Perceptron Graph Neural Network (GVP-GNN) (Jing et al., 2021) encoder.

**Featurization**  We represent each RNA tertiary structure as a geometric graph $\mathcal{G} = (\mathcal{V}, \mathcal{E})$, where $\mathcal{V}$ denotes nucleotides as nodes and $\mathcal{E}$ encodes spatial adjacencies. Each nucleotide $i$ is characterized

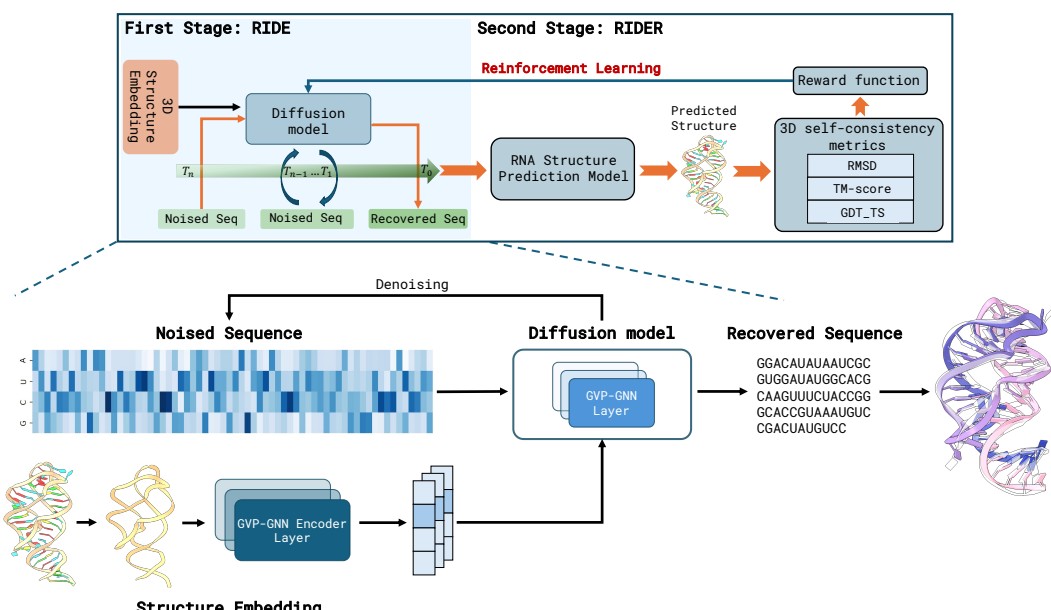

Figure 6: The proposed framework consists of two stages: RIDE (pre-training via supervised learning) and RIDER (fine-tuning via reinforcement learning). A multi-layer GVP-GNN encoder-decoder backbone is used to iteratively denoise RNA sequences conditioned on the target 3D structure. Structural fidelity is evaluated using a separate RNA structure predictor and fed back through a reward function.

by the coordinates of three backbone atoms: P, C4', and N1 (for pyrimidines) or N9 (for purines). The node coordinate $\vec{x}_i \in \mathbb{R}^3$ is defined as the centroid of these atoms. To capture local geometry, edges connect each node to its $k = 32$ nearest spatial neighbors.

Node features $h_i^{(0)}$ consist of scalar features $s_i \in \mathbb{R}^{d_s^{in}}$ and vector features $\vec{v}_i \in \mathbb{R}^{d_v^{in} \times 3}$. Scalar features include intra-nucleotide distances and angles, while vector features capture backbone orientation, such as the unit vector $\vec{x}_{i+1} - \vec{x}_i$ and vectors from C4' to P and N1/N9. Edge features $e_{ij}^{(0)}$ for an edge between node $i$ and $j$ similarly consist of scalar components $s_{ij} \in \mathbb{R}^{d_{es}^{in}}$ (e.g., 3D distance $||\vec{x}_j - \vec{x}_i||_2$ encoded via radial basis functions, and sequence separation $j - i$ encoded via sinusoidal positional encodings) and vector components $\vec{e}_{ij} \in \mathbb{R}^{d_{ev}^{in} \times 3}$ (e.g., the unit vector $\vec{x}_j - \vec{x}_i$).

**Structure encoder**    The featurized graph is processed by a structure encoder composed of $L_E = 5$ GVP-GNN layers. Each layer $l$ updates the node representations $(s_i^{(l)}, \vec{v}_i^{(l)})$ by aggregating information from neighboring nodes and edges in an $E(3)$-equivariant manner. The update rule for a node $i$ at layer $l$ can be expressed as:

$$m_i^{(l)}, \vec{m}_i^{(l)} = \underset{j \in \mathcal{N}(i)}{\mathrm{AGG}} \left( \phi_{\mathrm{msg}}^{(l)} \left( (s_i^{(l-1)}, \vec{v}_i^{(l-1)}), (s_j^{(l-1)}, \vec{v}_j^{(l-1)}), (s_{ij}, \vec{e}_{ij}) \right) \right), \tag{14}$$

$$s_i^{(l)}, \vec{v}_i^{(l)} = \phi_{\mathrm{update}}^{(l)} \left( (s_i^{(l-1)}, \vec{v}_i^{(l-1)}), (m_i^{(l)}, \vec{m}_i^{(l)}) \right), \tag{15}$$

where $\mathcal{N}(i)$ denotes the neighbors of node $i$, $\phi_{\mathrm{msg}}^{(l)}$ and $\phi_{\mathrm{update}}^{(l)}$ are GVP-based message and update functions respectively, and AGG is an aggregation operator (e.g., mean). The encoder network first embeds the initial node and edge features using GVP layers $W_v$ and $W_e$:

$$(s_i^{(0)}, \vec{v}_i^{(0)}) = W_v((s_i^{in}, \vec{v}_i^{in})), \quad (s_{ij}^{(0)}, \vec{e}_{ij}^{(0)}) = W_e((s_{ij}^{in}, \vec{e}_{ij}^{in})). \tag{16}$$

The final output of the structure encoder is a set of node-level conditional embeddings $h_c = \{(s_i^{(L_E)}, \vec{v}_i^{(L_E)})\}$ for all $i \in \mathcal{V}$. These embeddings, $h_c$, encapsulate the essential 3D structural information and serve as the conditioning context for the diffusion model.

### B.2 RIDE: CONDITIONAL DIFFUSION MODEL ARCHITECTURE

The RIDE model is a conditional diffusion network built upon Geometric Vector Perceptron Graph Neural Networks (GVP-GNNs), specifically designed for RNA inverse folding.

**GVP-GNN configuration.** Both the structure encoder and the noise prediction network within the diffusion model comprise $L_E = L_D = 5$ layers of GVP-GNNs. The key architectural parameters and node/edge feature dimensions are summarized in Table 3. A dropout rate of $0.5$ is applied to each GVP-GNN layer to mitigate overfitting. To enhance robustness, Gaussian noise with a standard deviation of $0.1$ is added to node coordinates during training. The model incorporates time-step embeddings as additional conditioning inputs to guide the denoising process. In total, RIDE contains $10,214,213$ trainable parameters.

Table 3: RIDE architecture and associated hyperparameters.

| Hyperparameter | Value |
|---|---|
| Number of GVP-GNN layers ($L_E$, $L_D$) | 5 |
| Node input dimensions (scalar, vector) | (15, 4) |
| Node hidden dimensions (scalar, vector) | (256, 24) |
| Edge input dimensions (scalar, vector) | (131, 3) |
| Edge hidden dimensions (scalar, vector) | (128, 4) |
| Dropout rate | 0.5 |
| Output dimension | 4 |
| Number of nearest neighbors per node | 32 |
| Number of radial basis functions (RBFs) | 32 |
| Positional encoding dimension for edges | 32 |

**Diffusion model pre-training.** RIDE is pre-trained via noise prediction using a Mean Squared Error (MSE) loss. The forward noising process follows a Variance-Preserving (VP) Stochastic Differential Equation (SDE). Key hyperparameters for the SDE and training schedule are summarized in Table 4. The initial learning rate is set to $3 \times 10^{-4}$ and is decayed by a factor of $0.9$ using a ReduceLROnPlateau scheduler when the validation performance (measured by NSR) fails to improve for 5 consecutive epochs.

### B.3 DETAILS OF RIDER

Following pre-training, the RIDE model is fine-tuned using an advantage-based policy gradient algorithm, referred to as RIDER.

**RL Algorithm Settings.** Reinforcement learning fine-tuning is performed using the Adam optimizer without any learning rate scheduling. Key hyperparameters for the training loop, policy updates, and experience collection are summarized in Table 5.

**Reward Function Scaling Factors.** The reward functions used during RL fine-tuning based on three structural similarity metrics. The scaling coefficients for these components, as defined in Section 3.3, are summarized in Table 6.

### B.4 TRAINING HARDWARE

All model pre-training and reinforcement learning fine-tuning experiments were conducted on servers equipped with NVIDIA RTX 4090 GPUs and Intel Core i9-13900K CPUs. With automatic mixed precision (AMP) enabled, training requires approximately 16 GB of GPU memory; without AMP, the memory usage increases to around 23 GB. RL fine-tuning requires approximately 8 hours on a single RTX 4090 GPU, while the critical inference cost remains unchanged and highly efficient.

Table 4: Training hyperparameters for `RIDE` pre-training.

| Hyperparameter | Value |
|---|---|
| SDE schedule | Linear |
| Initial noise scale $\beta_0$ | 0.1 |
| Final noise scale $\beta_1$ | 20.0 |
| Total diffusion time $T$ | 1.0 |
| Minimum time step $\epsilon_{\texttt{time}}$ | 0.001 |
| Number of epochs | 150 |
| Initial learning rate | $3 \times 10^{-4}$ |
| Learning rate scheduler | `ReduceLROnPlateau` |
| Scheduler decay factor | 0.9 |
| Scheduler patience | 5 epochs |
| Optimizer | `Adam` |
| Activation function | `SiLU` |
| Max nodes per batch | 3000 |
| Max nodes per sample | 5000 |
| DDIM denoising steps $N_{\texttt{steps}}$ | 50 |
| Sampling temperature | 0.1 |

Table 5: Training hyperparameters for `RIDER` fine-tuning.

| Hyperparameter | Value |
|---|---|
| Learning rate | $5 \times 10^{-5}$ |
| Optimizer | Adam |
| Number of RL training epochs | 80 |
| Policy updates per epoch | 2 |
| Gradient accumulation steps | 60 |
| Clip range $\epsilon_{\texttt{clip}}$ | 0.5 |
| Max gradient norm | 1.0 |
| Batch size | 60 |
| DDIM denoising steps during RL | 30 |

Table 6: Scaling factors for reward function components.

| Component | Scaling Factor |
|---|---|
| GDT_TS weight ($w_{\texttt{gdt\_scale}}$) | 5 |
| TM-score weight ($w_{\texttt{tm\_scale}}$) | 5 |
| RMSD weight ($w_{\texttt{rmsd\_scale}}$) | 0.5 |
| GDT_TS weight for bonus reward ($w_{\texttt{bonus\_gdt}}$) | 100 |
| RMSD weight for bonus reward ($w_{\texttt{bonus\_rmsd}}$) | 20 |

## C  3D STRUCTURE SELF-CONSISTENCY METRICS

To evaluate how accurately a designed RNA sequence folds back into its intended 3D backbone structure, we adopt three complementary global metrics: Root Mean Square Deviation (RMSD),

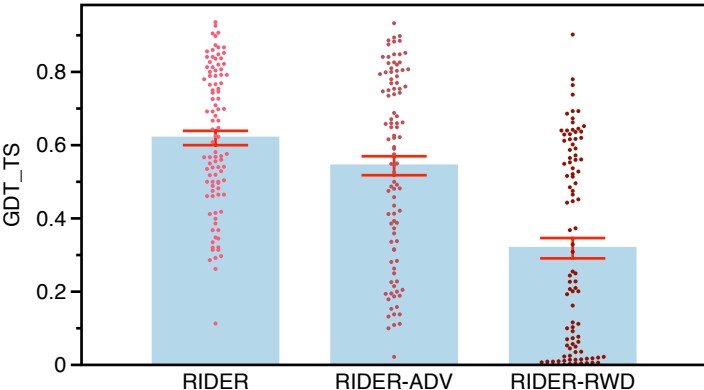

Figure 7: Comparison of `RIDER`, `RIDER-RWD`, and `RIDER-ADV` on the GDT_TS metric across the test set. Bar heights represent the mean, and red error bars indicate the Standard Error of the Mean (SEM).

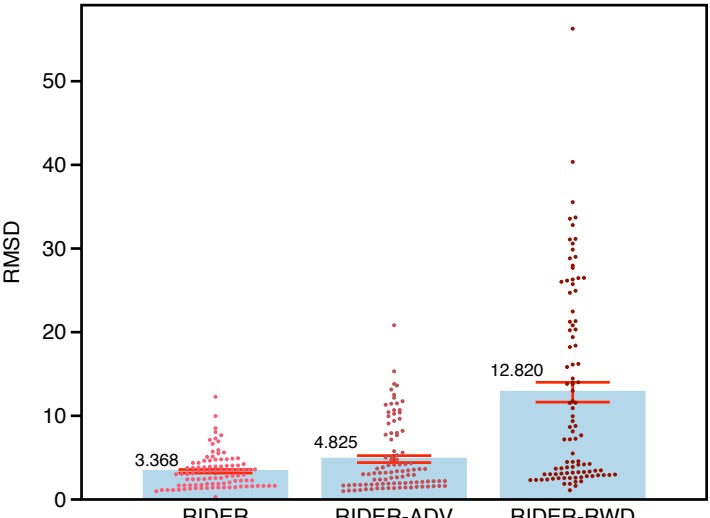

Figure 8: Comparison of `RIDER`, `RIDER-RWD`, and `RIDER-ADV` on the RMSD metric across the test set. Bar heights represent the mean, and red error bars indicate the SEM.

Global Distance Test Total Score (GDT_TS), and Template Modeling Score (TM-score). These metrics capture distinct aspects of structural similarity and are defined as follows.

## C.1 ROOT-MEAN-SQUARE DEVIATION (RMSD)

RMSD quantifies the average atomic displacement between two structures after optimal superposition. Given two sets of $N$ equivalent atoms with coordinates $\{\mathbf{x}_i\}$ (predicted) and $\{\mathbf{y}_i\}$ (reference), RMSD is defined as:

$$\text{RMSD} = \sqrt{\frac{1}{N} \sum_{i=1}^{N} \left\| \mathbf{x}_i - \mathbf{y}_i \right\|^2}. \tag{17}$$

Lower RMSD values indicate closer structural alignment. High-accuracy models typically achieve RMSD $< 2$Å, while poorly modeled regions may exceed 10Å.

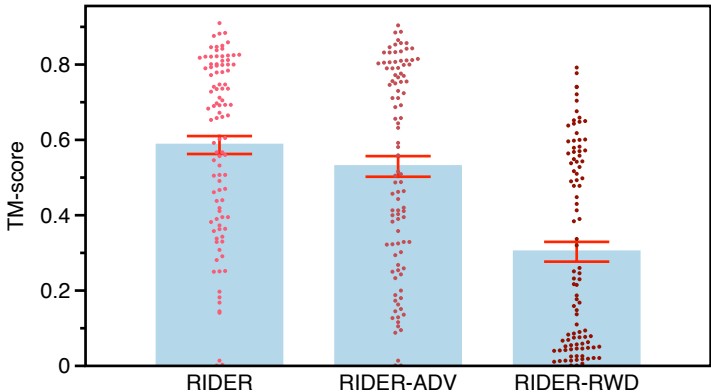

Figure 9: Comparison of `RIDER`, `RIDER-RWD`, and `RIDER-ADV` on the TM-score metric across the test set. Bar heights represent the mean, and red error bars indicate the SEM.

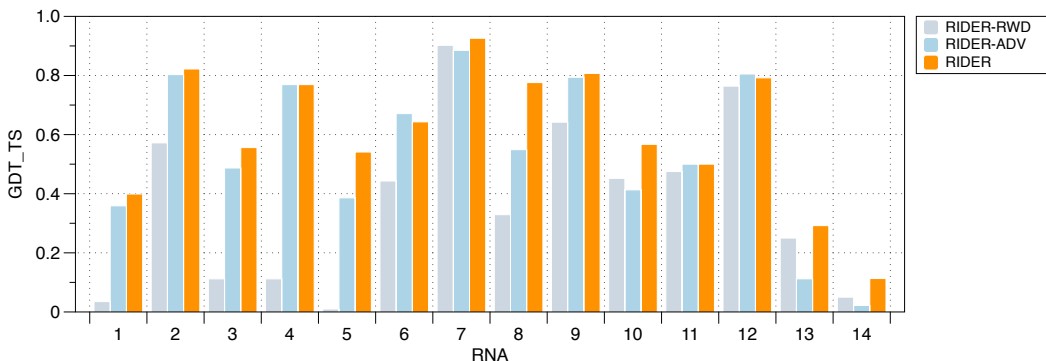

Figure 10: GDT_TS scores of `RIDER`, `RIDER-RWD`, and `RIDER-ADV` on 14 target RNA structures.

## C.2 GLOBAL DISTANCE TEST TOTAL SCORE (GDT_TS)

GDT_TS quantifies the fraction of residues that can be superimposed within multiple distance cut-offs. For a reference structure of length $N$, let $N_{d_k}$ denote the number of C$\alpha$ atom pairs within a distance threshold $d_k \in \{1, 2, 4, 8\}$Å, computed under independently optimized superpositions for each $d_k$. GDT_TS is then defined as:

$$\text{GDT\_TS} = \frac{1}{4N} \sum_{k \in \{1,2,4,8\}} N_{d_k} \times 100\%. \tag{18}$$

By focusing only on well-aligned residues ($d_k \leq 8$Å), GDT_TS is robust to local distortions and emphasizes the proportion of accurately modeled structure.

## C.3 TEMPLATE MODELING SCORE (TM-SCORE)

TM-score evaluates global fold similarity using a single optimized superposition with a length-dependent normalization term. Given $L_N$ residues in the reference structure and $L_T$ aligned residue pairs with pairwise distances $d_i$, the TM-score is defined as:

$$\text{TM-score} = \max \left[ \frac{1}{L_N} \sum_{i=1}^{L_T} \frac{1}{1 + \left( \frac{d_i}{d_0(L_N)} \right)^2} \right], \tag{19}$$

where the normalization scale $d_0$ is a function of $L_N$:

$$d_0(L_N) = 1.24 \sqrt[3]{L_N - 15} - 1.8. \tag{20}$$

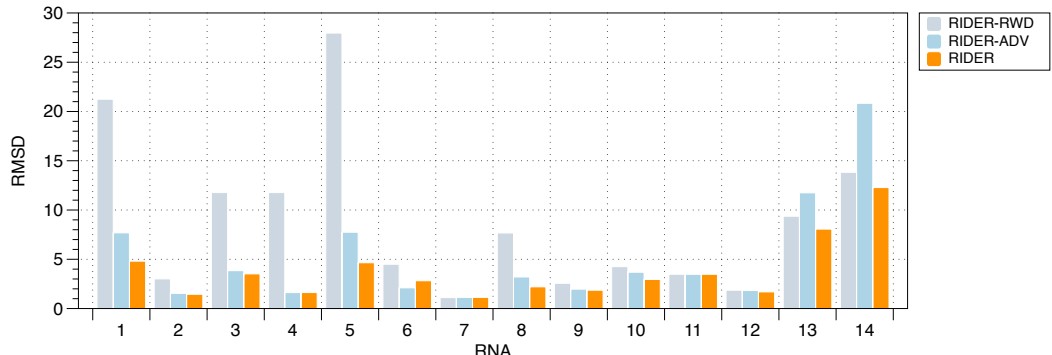

Figure 11: RMSD scores of `RIDER`, `RIDER-RWD`, and `RIDER-ADV` on 14 target RNA structures. Lower values indicate better structural alignment.

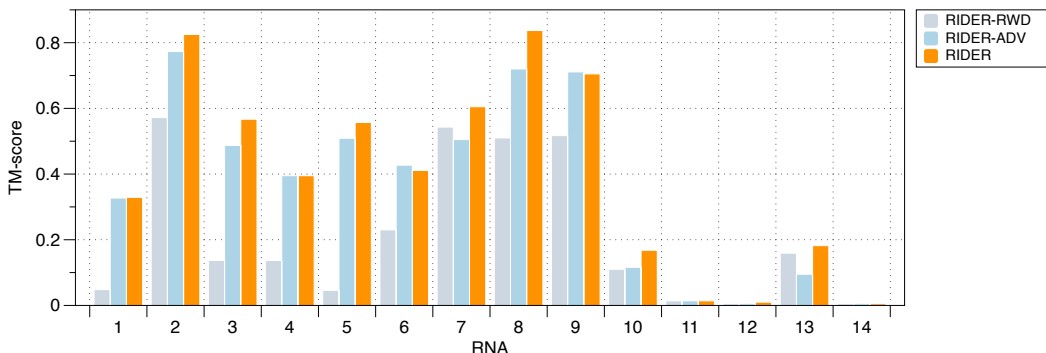

Figure 12: TM-score values of `RIDER`, `RIDER-RWD`, and `RIDER-ADV` on 14 target RNA structures. Higher values indicate better structural similarity.

This length correction makes TM-score more stringent for short structures (smaller $d_0$) and more permissive for longer ones, enabling fair comparison across proteins or RNAs of varying lengths.

### C.4    COMPARISON OF METRICS

RMSD provides a direct measure of average atomic deviation and is highly sensitive to outliers. GDT_TS quantifies the percentage of residues that fall within multiple distance thresholds, emphasizing well-modeled regions. TM-score yields a length-normalized global similarity score, which is less sensitive to local deviations and more suitable for comparing structures of different sizes. Together, these three metrics offer a comprehensive evaluation framework for assessing 3D structural fidelity in RNA inverse design.

## D    ABLATION STUDY

We conduct ablation studies on the advantage estimation strategy, reward design, and key hyperparameters used in both `RIDER` and `RIDE`.

### D.1    ADVANTAGE ESTIMATION STRATEGY

In `RIDER`, we replace raw rewards with advantage estimates and use the batch mean as a baseline for advantage calculation. Additionally, we introduce a moving average strategy to stabilize the baseline and reduce variance during training. In this section, we compare the full `RIDER` with two variants: (1) **RIDER-RWD**, which directly uses raw rewards without advantage estimation; and (2) **RIDER-ADV**, which uses batch-based advantage but without the moving average baseline.

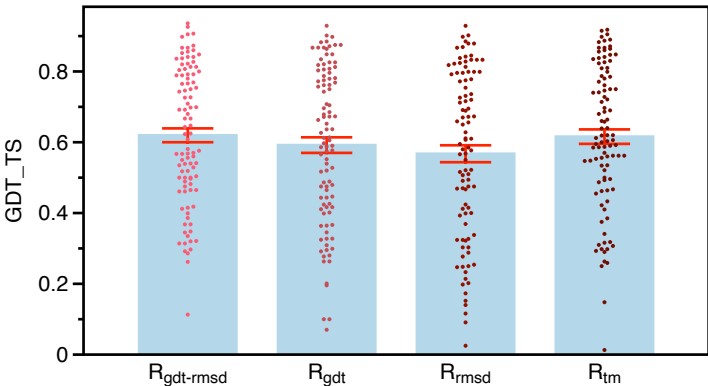

Figure 13: Comparison of four reward functions ($R^{\mathrm{tm}}$, $R^{\mathrm{rmsd}}$, $R^{\mathrm{gdt}}$, $R^{\mathrm{gdt\_rmsd}}$) on the GDT_TS metric across the test set. Bar heights represent the mean, and red error bars indicate the SEM.

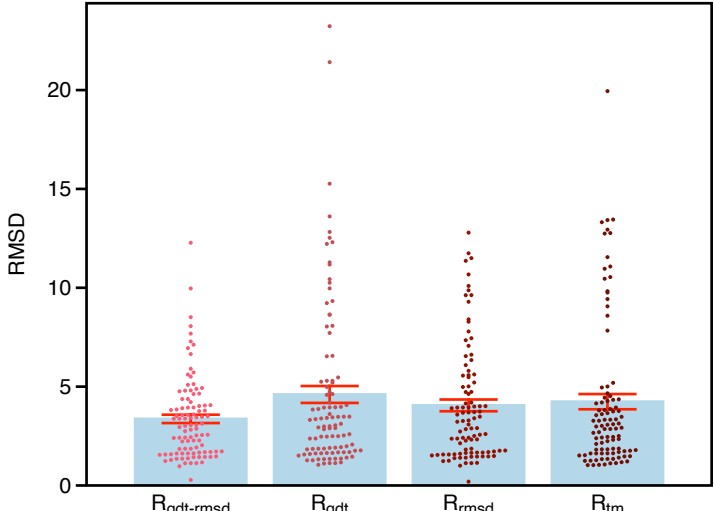

Figure 14: Comparison of four reward functions on the RMSD metric across the test set. Bar heights represent the mean, and red error bars indicate the SEM.

Figures 7, 8, and 9 show the performance of the three methods on the test set across three 3D self-consistency metrics. RIDER consistently achieves the best performance across all metrics. In contrast, RIDER-RWD performs poorly on many samples—for example, exhibiting numerous points with GDT_TS scores below 0.1 or RMSD values exceeding 10Å. This instability is likely due to the lack of advantage estimation, which causes high variance and sometimes leads to training collapse in later stages.

Compared to RIDER-ADV, our full RIDER achieves around a 15% improvement in GDT_TS, demonstrating the effectiveness of the moving average strategy in stabilizing learning and enhancing structural quality.

Figures 10, 11, and 12 further present the evaluation results of RIDER, RIDER-RWD, and RIDER-ADV on the 14 RNA structures of interest Das et al. (2010). On the GDT_TS metric, RIDER achieves the best performance across 12 out of the 14 samples, and it consistently outperforms the other variants across all three evaluation metrics. For the remaining two samples, RIDER performs slightly worse than RIDER-ADV but still significantly outperforms RIDER-RWD.

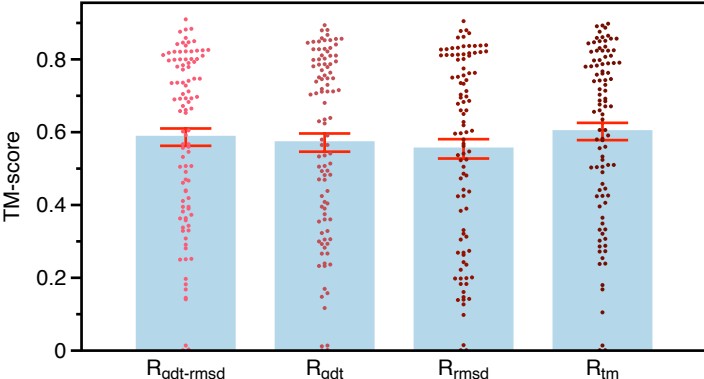

Figure 15: Comparison of four reward functions on the TM-score metric across the test set. Bar heights represent the mean, and red error bars indicate the SEM.

## D.2 REWARD FUNCTIONS

Figures 13, 14, and 15 present the evaluation results of the four reward functions we designed, measured across all three structural similarity metrics on the test set.

Among them, the reward function based solely on TM-score, $R^{\text{tm}}$, performs well on both TM-score and GDT_TS, but slightly worse on RMSD. The variants using only $R^{\text{rmsd}}$ or $R^{\text{gdt}}$ perform slightly worse than $R^{\text{tm}}$ overall.

We hypothesize that the inferior performance of $R^{\text{gdt}}$ may be due to the fact that GDT_TS only considers well-modeled residues (i.e., those within 8Å), and ignores errors in poorly modeled regions. This may cause the RL optimization to become trapped in local optima. On the other hand, while RMSD accounts for deviations across all residues, it does not apply length normalization like TM-score, making it potentially unsuitable for comparing RNAs of varying lengths. Moreover, in the course of exploration, some structures may yield very large RMSD values, leading to abrupt changes in reward, which can destabilize the training process.

In contrast, the composite reward $R^{\text{gdt-rmsd}}$, which combines both GDT_TS and RMSD, demonstrates strong performance across all metrics. We attribute this to the complementary nature of the two metrics—balancing local accuracy (GDT_TS) with global deviation sensitivity (RMSD).

## D.3 HYPERPARAMETER SENSITIVITY OF RIDE

We also perform a hyperparameter sensitivity analysis for the RIDE model. Table 7 summarizes the native sequence recovery performance under different configurations of optimizer, activation function, initial learning rate, and the number of GVP-GNN layers ($L_E, L_D$). The results are averaged over five independent runs with different random seeds.

The results show that the setting used in our main experiments—Adam optimizer, SiLU activation, initial learning rate of $3 \times 10^{-4}$, and 5 GVP-GNN layers—achieves the best sequence recovery ($0.61 \pm 0.02$). Deeper networks (8 layers) tend to underperform, possibly due to overfitting or optimization instability. We also observe that SiLU consistently outperforms ReLU, and that AdamW performs slightly worse than Adam under most configurations.

## D.4 ADDITIONAL RESULTS

Figure 4 in Section 4 presents the GDT_TS comparison on 14 RNA structures of interest Das et al. (2010) for gRNAde, RIDE (pre-trained), and RIDER (fine-tuned with $R^{\text{gdt-rmsd}}$). Here, we provide the corresponding supplementary results on the other two metrics: RMSD and TM-score.

Figure 16 shows the RMSD comparison among the three methods, while Figure 17 presents the TM-score comparison. These results further demonstrate that RIDER substantially outperforms both

Table 7: Ablation study of RIDE hyperparameters on native sequence recovery. Results are reported as the mean and standard deviation over five independent runs.

| Optimizer | Activation function | Initial LR | GVP-GNN layers $(L_E, L_D)$ | Native sequence recovery |
|---|---|---|---|---|
| Adam | SiLU | $3 \times 10^{-4}$ | 5 | **0.61 ± 0.02** |
| | | | 8 | 0.58 ± 0.04 |
| | | $5 \times 10^{-4}$ | 5 | 0.60 ± 0.02 |
| | | | 8 | 0.52 ± 0.08 |
| | ReLU | $3 \times 10^{-4}$ | 5 | 0.58 ± 0.04 |
| | | | 8 | 0.59 ± 0.03 |
| | | $5 \times 10^{-4}$ | 5 | 0.57 ± 0.03 |
| | | | 8 | 0.51 ± 0.04 |
| AdamW | SiLU | $3 \times 10^{-4}$ | 5 | 0.59 ± 0.03 |
| | | | 8 | 0.59 ± 0.04 |
| | | $5 \times 10^{-4}$ | 5 | 0.47 ± 0.10 |
| | | | 8 | 0.50 ± 0.08 |
| | ReLU | $3 \times 10^{-4}$ | 5 | 0.56 ± 0.03 |
| | | | 8 | 0.53 ± 0.09 |
| | | $5 \times 10^{-4}$ | 5 | 0.56 ± 0.04 |
| | | | 8 | 0.50 ± 0.10 |

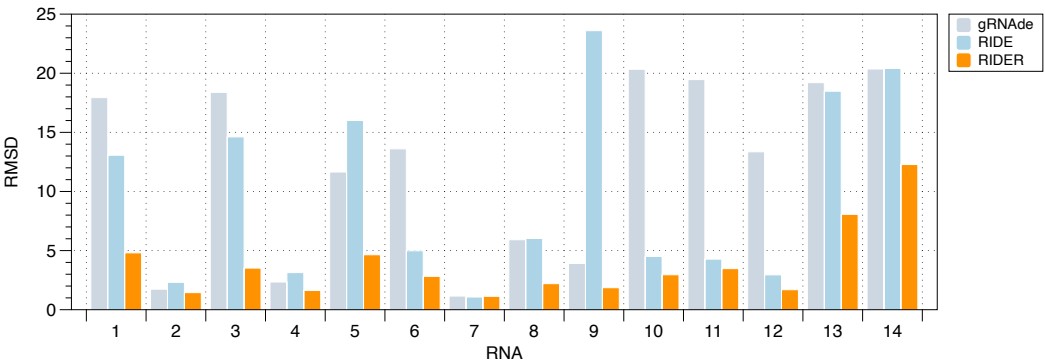

Figure 16: RMSD comparison of gRNAde, RIDE, and RIDER on 14 RNA structures of interest. Lower RMSD indicates better structural alignment.

gRNAde and the pre-trained RIDE on the RMSD and TM-score metrics. In most cases, RIDER achieves improvements exceeding $100\%$, highlighting the effectiveness of our proposed method.

## E DISCUSSION

### E.1 FUTURE WORK & LIMITATIONS

The proposed RIDER framework is flexible and can be extended in several directions. Future work may explore the design of multi-objective reward functions that incorporate additional factors such as sequence diversity or functional properties. Another promising avenue is to integrate biophysical metrics, for example, energy-based terms, into the reward function. Such integration could guide the model towards sequences with improved thermodynamic stability, which is a critical requirement for practical applications.

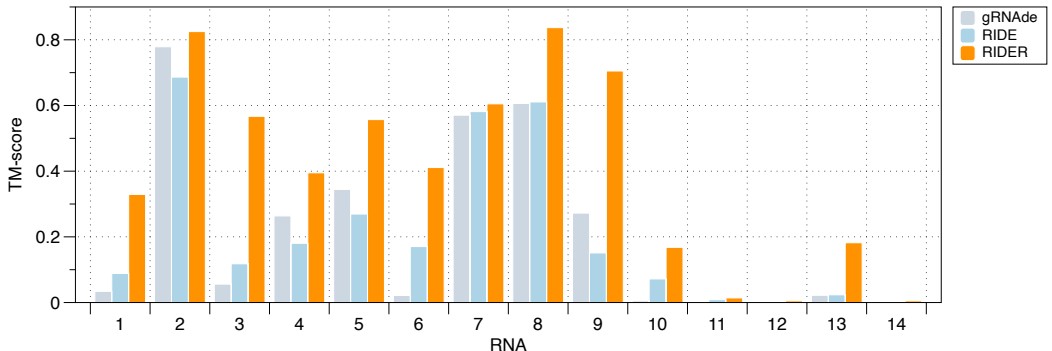

Figure 17: TM-score comparison of gRNAde, `RIDE`, and `RIDER` on 14 RNA structures of interest. Higher TM-score reflects better global fold similarity.

Our current method focuses on *in silico* RNA inverse design. However, RNA molecules are dynamic and can adopt multiple conformations *in vivo*, further increasing the complexity of the inverse design task. As future work, we plan to experimentally validate our designed sequences and extend the framework to support multiple, potentially competing, optimization objectives.

## E.2 BROADER IMPACTS

The proposed method may accelerate progress in synthetic biology, RNA-based drug development, and the study of gene regulatory mechanisms, thereby contributing to novel therapeutic strategies. By enabling the design of RNAs with precise 3D structures, it opens opportunities for engineering functional RNA circuits, aptamers, and ribozymes. However, applying AI models to biological sequence design introduces challenges related to interpretability and reliability. Enhancing the interpretability of such models—e.g., by identifying which structural features influence predictions—may help mitigate these concerns and promote responsible use in biological design.

