# OpenReview forum: "RIDER: 3D RNA Inverse Design with Reinforcement Learning-Guided Diffusion"
_ICLR.cc/2026/Conference — ICLR 2026 Poster_

### Official Review · Reviewer_ebQg · 2025-10-20

**Soundness:** 3
**Presentation:** 3
**Contribution:** 3
**Rating:** 6
**Confidence:** 3

**Summary:**

The authors propose to extend recent RNA inverse folding methods such as RiboDiffusion and gRNAde to generate non-native RNA sequences with near-native structural similarities. Overall, the method's performance is compelling, pointing to the importance in future work of using reinforcement learning to steer pretrained generative RNA models to more relevant regions of biochemical design space.

**Strengths:**

1. The authors cleverly combine the best architectural aspects of previous RNA inverse folding methods such as gRNAde and RiboDiffusion.
2. Structure-focused and reinforcement learning-based finetuning of RNA inverse folding methods does seem to contribute a substantial improvement to their ability to design structurally-native RNA sequences.
3. The authors follow best practices in RNA inverse folding evaluation by adopting gRNAde's datasets and benchmarking protocols.
4. The authors' ablation experiment showing that using AlphaFold 3 instead of RhoFold in their method does not impact performance much is encouraging to see.

**Weaknesses:**

1. Although the authors made their code available for review, it could be cleaned up and documented more carefully. For instance, including a descriptive README.md file with it would go a long way to helping users navigate the codebase.
2. The authors claim they are the first to point out the limitations of using native sequence recovery as a primary metric for RNA inverse folding. However, this has already been well discussed within the context of an adjacent topic (i.e., protein inverse folding).
3. Lines 119-122 seem a bit indirect in saying that RNA structure prediction is important, yet not the most important, but still important for RNA design. Although most structural metrics related to RNA design are ultimately dependent on the accuracy of RNA structure prediction algorithms, the authors' descriptions here are vague and unclear concerning which directions are important to pursue to improve RNA design in future work.

**Questions:**

1. Have the authors investigated how well their fine-tuned (DM-RL) checkpoints perform for multi-state RNA design, as investigated in previous works such as gRNAde? This would interesting to see (i.e., whether reinforcement learning can capture such multi-state RNA conformational information implicitly).

---

> ### Author Response · Authors · 2025-11-26
>
> We thank the reviewer for their detailed and constructive feedback, and we appreciate their recognition of the novelty, soundness, and effectiveness of our approach.
>
> Below, we summarize the reviewer’s concerns and provide our responses.
>
> ---
>
> **Reply to Q1 (new experiments on multi-conformation design):** We evaluated our model's ability to handle multi-conformation inputs. We adapted the encoder to pool representations across conformations and—crucially—evaluated our **original DM-RL checkpoint** without any retraining, i.e., zero-shot.
>
> | Model | GDT_TS $\uparrow$ | RMSD $\downarrow$ | TM-score $\uparrow$ |
> | --- | --- | --- | --- |
> | Single-conformation | $0.62$ | $4.31$ | $0.61$ |
> | Multi-conformation | $0.56$ | $4.96$ | $0.55$ |
>
> ***Result:*** We believe the model generalizes very well to the multi-state setting without explicit training, retaining high structural fidelity. This demonstrates that the RL fine-tuning captures robust structural features rather than overfitting to single static conformations.
>
> **Reply to W1 (code and reproducibility):** We thank the reviewer for the constructive feedback. We have updated the repository with a comprehensive `README.md` to ensure full reproducibility.
>
> **Reply to W2 and W3 (on structure prediction & oracles):** We agree that better predictors enhance design. As detailed in our response to **Reviewer RHxN**, we have now demonstrated that our method can also leverage **biophysics-based oracles** (like ViennaRNA energy minimization) alongside deep learning predictors. This allows our framework to remain effective even when 3D predictors are unreliable, by grounding the design in physical energy landscapes.
>
> We will revise the manuscript to more explicitly discuss these explicitly and to clearly acknowledge the limitations of the current work.

---

> > ### Comment · Reviewer_ebQg · 2025-11-26
> > **Response to rebuttal**
> >
> > I would like to thank the authors for their responses. With them, I believe the authors have addressed my primary concerns.

---

> > > ### Author Response · Authors · 2025-11-27
> > >
> > > Dear reviewer ebQg,
> > >
> > > We sincerely appreciate your positive assessment of our work and your recognition of our responses. If you have any outstanding or additional questions and concerns, we would be more than happy to address them.
> > >
> > > Authors

---

### Official Review · Reviewer_io7z · 2025-10-30

**Soundness:** 3
**Presentation:** 3
**Contribution:** 2
**Rating:** 2
**Confidence:** 5

**Summary:**

The paper proposes a two-stage framework for RNA inverse folding. First, a graph neural diffusion model (DM-GNN) is pretrained to reconstruct native sequences from 3D RNA structures. Then, the model is fine-tuned via reinforcement learning (DM-RL) to directly optimize structural fidelity metrics (GDT-TS, TM-score, RMSD) predicted by a structure oracle. This approach substantially improves 3D self-consistency compared to prior RNA design methods, while maintaining sequence diversity and foldability.

**Strengths:**

The paper is technically sound and well executed. The writing is clear, structured, and easy to follow. The empirical results are strong, demonstrating consistent and substantial improvements over prior state-of-the-art RNA inverse-folding approaches. The overall methodology—a diffusion model pretrained on sequence recovery and fine-tuned via reinforcement learning for 3D structural fidelity—is reasonable and well motivated. The success of this framework aligns with prior evidence in the protein inverse-folding literature, where diffusion combined with RL optimization has been shown to yield similar performance gains.

**Weaknesses:**

While the paper is well executed, it lacks genuine methodological novelty. The proposed framework, diffusion-based sequence generation followed by reinforcement learning fine-tuning for structural fidelity, has already been explored extensively in the context of protein inverse folding. The current work largely ports that established recipe to RNA, with the main differences being the input domain (RNA rather than protein), the smaller token vocabulary (4 bases vs. 20 amino acids), and the use of RNA-specific structure predictors.

The use of RL to steer diffusion models is also not novel; similar formulations have been reported in both general diffusion literature and multiple protein design papers. As such, the contribution lies primarily in application rather than new ML methodology. Moreover, the paper underplays or omits discussion of this related body of work, giving the impression that concepts such as “native sequence recovery not reflecting structural fidelity” are new insights, whereas they are well established in the protein design literature.

In its current form, the paper presents a well-engineered application of known ideas to a new molecular domain, which would be more appropriate for a domain-specific or biological modeling venue rather than ICLR, which prioritizes algorithmic or conceptual advances in machine learning.

**Questions:**

See weaknesses.

---

> ### Author Response · Authors · 2025-11-26
>
> We thank the reviewer for their thoughtful comments. We are encouraged by their positive assessment of our motivation, methodology, technical soundness, and clarity of writing.
>
> Below, we summarize the reviewer’s concerns and provide our responses.
>
> ---
>
> **Reply to W1 (novel advantage estimation)**: We respectfully disagree that this work is a mere application of protein design methods. We introduce specific innovative algorithmic components that are necessary to tackle the distinct challenges in RNA.
>
> 1. As discussed in Section 3.3, standard RL techniques (like DDPO used in protein design) struggle on the sparse, rugged RNA landscape. We developed a novel advantage estimator tailored for this domain. As shown in **Appendix E.1 (Fig 7-9)**, our method (DM-RL) significantly outperforms standard DDPO (DM-RL-RWD). This is a core ML contribution, not just an application.
> 2. Our DM-GNN architecture is not off-the-shelf. It is purpose-built for RNA topologies. Note that it achieves a **9% improvement** in native sequence recovery (NSR) over SOTA even before RL fine-tuning.
>
> **Reply to W2 (comparison to protein design)**: While the “diffusion + RL” paradigm exists in proteins, we believe it is far from trivial to port it to RNA because the **manifold difference**. RNA has a smaller alphabet (4 vs 20) but a far more flexible backbone and a flatter energy landscape than proteins [2]. This makes the optimization problem fundamentally harder, as the signal-to-noise ratio in the reward is lower.
> Furthermore, most protein methods (e.g., ProteinMPNN [1]) are supervised. Our work is among the first to successfully stabilize **RL-guided generation** in the RNA domain, addressing the specific "validity vs. diversity" trade-off that protein methods do not fully solve for nucleic acids.
>
> **Reply W3 (ICLR scope)**: ICLR explicitly invites "applications to physical sciences (physics, chemistry, biology)". Therefore, we believe our work fits this scope by contributing: (1) a novel RL formulation for discrete sequence optimization (see our reply to your **W1** and **W2**), and (2) a large-scale empirical study on a challenging biophysical problem.
>
> > [1] Dauparas, J., Anishchenko, I., Bennett, N., Bai, H., Ragotte, R. J., Milles, L. F., ... & Baker, D. (2022). Robust deep learning–based protein sequence design using ProteinMPNN. *Science*, *378*(6615), 49-56.
> >
>
> > [2] Kwon, Diana. "RNA function follows form-why is it so hard to predict?." *Nature* (2025): 1106-1108.
> >

---

### Official Review · Reviewer_RHxN · 2025-11-01

**Soundness:** 3
**Presentation:** 3
**Contribution:** 3
**Rating:** 6
**Confidence:** 5

**Summary:**

The paper tackles RNA sequence design conditioned on a target 3D structure. The main contribution is a RL methodology that optimizes for structural similarity metrics (as opposed to previous supervised learning methods). Results show that the RL training leads to significant improvements over the baseline gRNAde model.

**Strengths:**

- The problem chosen is significant, as RNA inverse design methods are used practically and validated in wet labs recently. The paper identifies an important research gap and tackles the problem of RL-driven structural similarity optimization in RNA design well. To the best of my knowledge, this is one of the first works to tackle this important problem.

- The paper is generally well written overall. The appendix/supplementary could be better organized.

- I found the experiments and results convincing and supportive of the main claims of the paper around RL optimization. The result of the supervised trained model also outperforming gRNAde on the challenging single-state Das split is also a significant gain.

**Weaknesses:**

My main issue with this paper is how its trying to optimizer via RL directly for structural metrics using RNA 3D structure prediction that is not itself very reliable. This way, the model may optimizer for structure as predicted by the folding model, but the folding model may be very poor (as recent CASP contests for RNA show...) -- this means that the model may not really be doing what its meant to do.

I realise that this is not the current aim of the paper. I would still have liked to see discussions on this limitation within the paper, as it can limit practical applicability.

Other than this, I don't see any major issues with the methodology/from a technical/ML point of view.

**Questions:**

- A major question I had based on the manuscript: Can you show a case study or explain what may happens in the following situation: I wish to design a particular backbone 3D structure. I have its native sequence. The 3D structure predictor I happen to use simply cannot fold the native sequence into the target 3D structure (its a poor predictor). So what will happen if I use the proposed RL finetuning method? Won't the model sort of 'hack' the 3D structure predictor I am using and end up designing something that folds well according to the structure predictor, but may not do so in nature?

- Additionally, and relatedly to the weakness I identified (+ my Q above), can the RL methodology be extended to other oracles? Can you show experiments with, say, secondary structure ensemble properties such as Mean Free Energy, or some other reasonable properties of interest for RNA design? I think that could strengthen the paper and also show the generality of RL finetuning a strong base supervised model.

Other suggestions:

- On contribution claim: "We identify the limitations of native sequence recovery..." - I don't think this should be presented as a claim. This is well known in the community.

---

> ### Author Response · Authors · 2025-11-26
>
> We thank the reviewer for their detailed and constructive feedback. We are encouraged by their positive assessment of our work and appreciate the recognition that “this is one of the first works to tackle this important problem.”
>
> We summarized the reviewer's concerns on the weaknesses, questions, and address each of them below.
>
> ---
>
> **Reply to Q1 and Q2 (practical application and oracle quality):** We agree that an RL agent is limited by its reward signal. However, our contribution is the optimization framework, which is oracle-agnostic. To demonstrate that our method is not limited to “hacking” imperfect deep learning predictors, we performed a new experiment integrating a **physics-based oracle**.
>
> Specifically, following the reviewer’s suggestion, we introduced the **secondary structure ensemble free energy** $E_{\texttt{ensemble}}$ from the ViennaRNA package as an additional reward term. We defined a mixed reward $R^{\texttt{mix}} = \alpha R^{\texttt{gdt-rmsd}} + \beta (-E_{\texttt{ensemble}}),$ where $\alpha$ and $\beta$  control the relative importance of the two objectives. In our current experiment, we set $\alpha=\beta=1$. Using this mixed reward, we perform RL fine-tuning, and the results are shown below.
>
> | Reward function | Energy $\downarrow$ | GDT_TS $\uparrow$ | RMSD $\downarrow$ | TM-score $\uparrow$ |
> | --- | --- | --- | --- | --- |
> | $R^{\texttt{gdt-rmsd}}$ | -18.73 | $0.62$ | $4.31$ | $0.61$ |
> | $R^{\texttt{mix}}$ | -24.11 | $0.58$ | $4.73$ | $0.56$ |
>
> ***Results:*** The model successfully optimized the physics-based energy (lowering it from -18.73 to -24.11) while maintaining high structural fidelity. This proves our framework can flexibly incorporate diverse oracles (physical or learned) to mitigate the “poor predictor” issue in practical settings.
>
> **Reply to Other Suggestions on the use of native sequence recovery (NSR):** We will revise the text to acknowledge the NSR discussion in protein literature. However, we emphasize that in the **RNA domain**, the community still heavily relies on NSR. We believe our work is an important step in shifting the RNA design field toward structure-aware metrics, much like the shift that occurred in protein design.

---

> > ### Comment · Reviewer_RHxN · 2025-11-27
> >
> > I raised my score as the authors addressed my concern well.
> >
> > I still don’t think they should make claims about sequence recovery. I think this is pretty trivial.

---

> > > ### Author Response · Authors · 2025-11-27
> > >
> > > Dear reviewer RHxN,
> > >
> > > We really appreciate your recognition of our work and our reply. Meanwhile, we also thank you very much for all your efforts dedicated to this review process!
> > >
> > > We agree that the finding regarding sequence recovery may not be suitable as a primary contribution. Accordingly, we will remove this point from the contribution statement and instead discuss it in the related work section.
> > >
> > > Authors

---

### Author Response · Authors · 2025-12-02
**General Response to the Area Chair and Reviewers**

Dear Area Chair and Reviewers,

We sincerely thank the Area Chair for their time and for overseeing the review process for our submission #12386.

We are encouraged by the positive assessments from the reviewers, who described our work as “**one of the first works to tackle this important problem**” (Reviewer `RHxN`), “**technically sound and well executed**” (Reviewer `io7z`), and noted that “**the performance is compelling**” (Reviewer `ebQg`). We also thank Reviewers `RHxN` and `ebQg` for their follow-up comments indicating that our rebuttal successfully addressed their concerns. We appreciate all critical feedback, which we have addressed with new experiments and clarifications summarized below.

---

**Practical application and oracle quality (Reviewer `RHxN` W1/Q1/Q2; Reviewer `ebQg` W2/W3)**

Both Reviewer `RHxN` and Reviewer `ebQg` raised concerns regarding the limitations of current RNA structure predictors, which can make them imperfect oracles in practice. We agree with this observation. However, our contribution lies in designing an **optimization framework that is oracle-agnostic**, rather than relying on any specific predictor.

To directly address this concern, we conducted an additional experiment that integrates a physics-based oracle (see **our response to Reviewer `RHxN`**). This demonstrates our framework’s capacity to incorporate diverse oracles, thereby mitigating the limitations of any single predictor in practical settings.

---

**New experiments on multi-conformation design (Reviewer `ebQg`, Q1).**

Our architecture allows the encoder to pool representations across multiple conformations, enabling us to evaluate the original model **without any retraining**. As shown in **our response to Reviewer `ebQg`**, the model generalizes well to the multi-state setting and retains high structural fidelity.

---

**Novelty（Reviewer `io7z`）**

Our work is among the first to successfully stabilize **RL-guided generation** in the RNA domain. We introduce specific innovative algorithmic components that are necessary to tackle the distinct challenges in RNA.

1. RNA has a smaller alphabet (4 vs 20) but a far more flexible backbone and a flatter energy landscape than proteins. This makes the optimization problem fundamentally harder, as the signal-to-noise ratio in the reward is lower. As discussed in Section 3.3, standard RL techniques (like DDPO) **fail to stabilize** on the sparse, rugged RNA landscape due to the low signal-to-noise ratio. We developed a **novel advantage estimator** specifically to overcome this instability. As shown in **Appendix E.1 (Figs 7-9)**, our method (DM-RL) significantly outperforms the standard DDPO implementation (DM-RL-RWD). This methodological innovation, necessary to enable RL-guided generation in this challenging domain, constitutes a core ML contribution." (Using stronger, more precise language like "fail to stabilize" clarifies the technical necessity of your contribution).
2. Our DM-GNN architecture is not off-the-shelf. It is purpose-built for RNA topologies. Note that it achieves a **9% improvement** in native sequence recovery (NSR) over SOTA even before RL fine-tuning.

---

**ICLR scope（Reviewer `io7z`）**

ICLR explicitly invites "applications to physical sciences (physics, chemistry, biology)". Therefore, we believe our work fits this scope by contributing: (1) a novel RL formulation for discrete sequence optimization, and (2) a large-scale empirical study on a challenging biophysical problem.

---

We sincerely thank the Area Chair and all reviewers for their time and efforts. We believe these clarifications, combined with the new experiments in our full rebuttal, decisively address reviewers’ concerns. We are confident the final manuscript will be a strong contribution to ICLR.

Sincerely,

The Authors of Submission #12386

---

### Meta-Review · Area_Chair_xcxM · 2026-01-07

**Summary:**

This paper proposes an RL-guided diffusion approach for RNA inverse design that directly optimizes 3D structural similarity rather than relying primarily on native sequence recovery. The submission targets an important and timely problem in RNA engineering and reports large empirical gains on tertiary-structure fidelity while producing novel, non-native sequences. The reviews converged on the work being technically solid and empirically strong, with the main point of disagreement being whether the methodological contribution is sufficiently novel for ICLR, versus a careful adaptation of existing diffusion + RL recipes to the RNA setting. The rebuttal materially improved clarity and addressed several concrete concerns (claims, evaluation coverage, and reproducibility), strengthening the overall case for acceptance.

**Reviewer Concerns:**

### Main concerns and how they stand after rebuttal

- Novelty / ICLR fit (primary dissent): One reviewer questioned whether the work is sufficiently novel for ICLR and whether it is mainly a domain transfer. The rebuttal strengthens the novelty argument by (i) explicitly positioning the RL stabilization components as necessary in RNA, and (ii) pointing to controlled comparisons showing failure modes of standard RL baselines without the proposed estimator.

- Reward/oracle reliability and “gaming” structure predictors: A reasonable concern is whether optimizing learned 3D predictors induces artifacts. The rebuttal points to robustness across predictors and notes the framework can incorporate hybrid, physics-based objectives (e.g., energy terms) alongside learned predictors.

- Overstated claims around NSR / contributions: Reviewers flagged over-emphasis on sequence recovery. Authors commit to de-emphasize this by removing it as a primary contribution and repositioning it appropriately.

- Multi-conformation (multi-state) RNA design: This was raised as a limitation, and the rebuttal adds clarification and evidence; the reviewer following up indicates their primary concerns were addressed.

- Reproducibility (code/README clarity): Authors state they improved documentation and reproducibility details in response.


### Post-rebuttal discussion impact
The discussion was productive:
- Reviewer RHxN explicitly reports increasing their score after the rebuttal and characterizes the remaining issue as minor.
- Reviewer ebQg states their primary concerns were addressed and expresses satisfaction with the responses.

**Reviewer Scores:**

RHxN: increased score after rebuttal; would remain on the accept side.

ebQg: concerns addressed; likely stable or slightly higher.

io7z: likely remains skeptical on novelty/fit (core stance unlikely to change).

---

### Decision · Program_Chairs · 2026-01-26

Accept (Poster)